# Inductive biases of multi-task learning and finetuning: multiple regimes of feature reuse

**Samuel Lippl**[*]
Center for Theoretical Neuroscience
Columbia University
New York, NY
samuel.lippl@columbia.edu

**Jack Lindsey**[*,**]
Anthropic
San Francisco, CA
jackwlindsey@gmail.com

## Abstract

Neural networks are often trained on multiple tasks, either simultaneously (multi-task learning, MTL) or sequentially (pretraining and subsequent finetuning, PT+FT). In particular, it is common practice to pretrain neural networks on a large auxiliary task before finetuning on a downstream task with fewer samples. Despite the prevalence of this approach, the inductive biases that arise from learning multiple tasks are poorly characterized. In this work, we address this gap. We describe novel implicit regularization penalties associated with MTL and PT+FT in diagonal linear networks and single-hidden-layer ReLU networks. These penalties indicate that MTL and PT+FT induce the network to reuse features in different ways. 1) Both MTL and PT+FT exhibit biases towards feature reuse between tasks, and towards sparsity in the set of learned features. We show a "conservation law" that implies a direct tradeoff between these two biases. 2) PT+FT exhibits a novel "nested feature selection" regime, not described by either the "lazy" or "rich" regimes identified in prior work, which biases it to *rely on a sparse subset* of the features learned during pretraining. This regime is much narrower for MTL. 3) PT+FT (but not MTL) in ReLU networks benefits from features that are correlated between the auxiliary and main task. We confirm these findings empirically with teacher-student models, and introduce a technique – weight rescaling following pretraining – that can elicit the nested feature selection regime. Finally, we validate our theory in deep neural networks trained on image classification. We find that weight rescaling improves performance when it causes models to display signatures of nested feature selection. Our results suggest that nested feature selection may be an important inductive bias for finetuning neural networks.

## 1 Introduction

Neural networks are often trained on multiple tasks, either simultaneously ("multi-task learning," henceforth MTL, see [1, 2]) or sequentially ("pretraining" and subsequent "finetuning," henceforth PT+FT, see [3, 4]). Empirically, models can transfer knowledge from auxiliary tasks to improve performance on tasks of interest. However, theoretical understanding of how auxiliary tasks influence learning and generalization is limited.

Auxiliary tasks are especially useful when there is less data available for the target task. Modern "foundation models," trained on data-rich general-purpose auxiliary tasks (like next-word prediction or image generation) before adaptation to downstream tasks, are a timely example of this use case [5].

---

[*]Equal contributions.

[**]Work primarily conducted while at the Center for Theoretical Neuroscience, Columbia University.

38th Conference on Neural Information Processing Systems (NeurIPS 2024).

Auxiliary tasks are also commonly used in reinforcement learning, where performance feedback can be scarce [6]. Intuitively, auxiliary task learning biases the target task solution to use representations shaped by the auxiliary task. When the tasks share common structure, this influence may enable generalization from relatively few training samples for the task of interest. However, it can also have downsides, causing a model to inherit undesirable biases from auxiliary task learning [7, 8].

An influential strategy in the literature on the theory of *single-task* learning has been to characterize the *implicit regularization* conferred by the combination of network architecture and optimization algorithm [9–13]. Alternatively, some others characterize the effects of *explicit parameter regularization* (e.g. an $\ell_2$-penalty on weights) on the inductive bias of networks towards learning certain functions [14–16]. Compared to the single-task case, the inductive bias of MTL and PT+FT, whether obtained via explicit regularization or implicit regularization induced by optimization dynamics, is less well understood. Here we make progress on this question by studying inductive biases of MTL and PT+FT in two network architectures that have been extensively theoretically studied: diagonal linear networks, and densely connected networks with one hidden layer and a ReLU nonlinearity. We then demonstrate that our insights transfer to practically relevant scenarios by studying deep neural networks trained on image classification tasks.

Our specific contributions are as follows:

- We characterize regularization penalties associated with MTL or PT+FT in both diagonal linear networks and single hidden-layer ReLU networks (Section 3).

- We find that both MTL and PT+FT are biased towards solutions that reuse features between tasks, and that rely on a sparse set of features (Section 4.2).

- We then find that under suitable scalings, PT+FT exhibits a "nested feature selection" regime, distinct from previously characterized "rich" and "lazy" regimes, which biases finetuning to extract sparse subsets of the features learned during pretraining (Sections 4.3 and 4.4).

- We find that PT+FT in ReLU networks can benefit from correlated (not just identical) features between the main and auxiliary task, but only when coefficients of features in the main task weights are of comparable magnitude (Section 4.5).

- Finally, we study deep neural networks trained on natural image data (CIFAR-100 and ImageNet) (Section 5). Remarkably, we find that rescaling weights before finetuning improves accuracy in ResNets. Our analysis of the network representations suggests that this weight rescaling also results in the network relying on a low-dimensional subspace of its pretrained representation (i.e. exhibiting nested feature selection behavior). Intriguingly, Vision Transformers already exhibit signatures of nested feature selection without weight rescaling, and do not benefit from weight rescaling, suggesting that the nested feature selection regime is beneficial for finetuning performance.

## 2 Related work

A variety of studies have characterized implicit regularization effects in deep learning. These include biases toward low-frequency functions [17], stable minima in the loss landscape [18], low-rank solutions [19], and lower-order moments of the data distribution [20]. Chizat & Bach [13] show that when using cross-entropy loss, shallow (single hidden layer) networks are biased to minimize the $\mathcal{F}_1$ norm, an infinite-dimensional analogue of the $\ell_1$ norm over the space of possible hidden-layer features [see also 12, 14]. Other work has shown that implicit regularization for mean squared error loss in nonlinear networks cannot be exactly characterized as norm minimization [21], though $\mathcal{F}_1$ norm minimization is a precise description under certain assumptions on the inputs [22].

Compared to the body of work on inductive biases of single-task learning, theoretical treatments of MTL and PT+FT are more scarce. Some prior studies have characterized benefits of multi-task learning with a shared representational layer in terms of bounds on sample efficiency [23–25]. Others have characterized the learning dynamics of deep linear networks trained from nonrandom initializations, which can be applied to understand finetuning dynamics [26, 27]. Similarly, insights on the implicit regularization of gradient descent in linear models has been applied to better understand forgetting and generalization in a continual learning setup [28–31]. However, while these works demonstrate an effect of pretrained initializations on learned solutions, the linear models they study do not capture the notion of feature learning we are interested in. Finally, teacher-student setups

have been used to study the impact of task similarity on continual learning in deep neural networks [32, 33]. This methodology could also be applied to our setup (i.e. to investigate generalization on a finetuning task), and, similarly, our tools could be applied to continual learning setups.

A few empirical studies have compared the performance of MTL vs. PT+FT in language tasks, with mixed results depending on the task studied [34, 35]. Others have observed that PT+FT outperforms PT + "linear probing" (training only the readout layer and keeping the previous layers frozen after pretraining), implying that finetuning benefits from the ability to learn task-specific features [36, 37].

**Inductive biases of diagonal linear networks.** The theoretical component of our study relies heavily on a line of work [38–42] that studies the inductive bias of a simplified "diagonal linear network" model. Diagonal linear networks parameterize linear maps $f : \mathbb{R}^D \to \mathbb{R}$ as

$$f_{\vec{w}}(\vec{x}) = \vec{\beta}(\vec{w}) \cdot \vec{x}, \qquad \beta_d(\vec{w}) := w^{(2)}_{+,d} w^{(1)}_{+,d} - w^{(2)}_{-,d} w^{(1)}_{-,d} \tag{1}$$

where $\vec{\beta}(\vec{w}) \in \mathbb{R}^D$. These correspond to two-layer linear networks in which the first layer consists of one-to-one connections, with duplicate $+$ and $-$ pathways to avoid saddle point dynamics around $\vec{w} = 0$. Woodworth *et al.* [38] showed that overparameterized diagonal linear networks trained with gradient descent on mean squared error loss find the zero-training-error solution that minimizes $\|\vec{\beta}\|_2$, when trained from large initial weight magnitude (the "lazy" regime, equivalent to ridge regression). When trained from small initial weight magnitude, networks instead minimize $\|\vec{\beta}\|_1$ (the "rich" regime). This bias is a linear analogue of feature learning/feature selection, as a model with an $\ell_1$ penalty tends to learn solutions that depend on a sparse set of input dimensions.

**Implicit vs. explicit regularization.** Theoretical work on the inductive biases conferred by different architectures has studied both the implicit regularization induced by gradient descent and explicit $\ell_2$-regularization. Notably, in homogeneous networks trained with crossentropy loss, implicit and explicit regularization yield identical inductive biases in the limit of infinite training time and infinitesimal regularization [12, 43], but this does not hold in general [38]. While [38, 40] are able to characterize the implicit regularization of gradient descent for diagonal linear networks, it is technically much more challenging to derive a similar result for multi-output diagonal linear networks, or for ReLU networks of any kind. In contrast, the impact of *explicit* weight regularization has been characterized for both multi-output diagonal linear networks [16] and multi-output ReLU networks [44–46].

Our main technical contributions to this theoretical landscape are (1) spelling out the implications of existing results on implicit regularization in diagonal linear networks and (2) providing a novel characterization of the inductive bias induced by applying explicit parameter regularization to the finetuning of ReLU networks from arbitrary initialization. Our choice to study explicit parameter regularization for ReLU networks is made primarily for theoretical tractability; we view these results as a proxy for understanding the more theoretically complex problem of implicit regularization.

# 3 Implicit and explicit regularization penalties for MTL and PT+FT

## 3.1 Theoretical setup

**Architectures.** First, we consider **diagonal linear networks** with hidden weights $\vec{w}_+, \vec{w}_- \in \mathbb{R}^D$ and $O \in \{1, 2\}$ output weights $v_+, v_- \in \mathbb{R}^{O \times D}$. ($O$ is 1 or 2 depending on the training paradigm, see below.) The resulting network function is defined as

$$f_{w,v}(\vec{x}) = \beta(w, v)\vec{x}, \quad \vec{\beta}_o(w, v) := \vec{v}_{+,o} \circ \vec{w}_+ - \vec{v}_{-,o} \circ \vec{w}_-, \quad \beta(w, v) \in \mathbb{R}^{O \times D}. \tag{2}$$

Second, we consider **ReLU networks** with $H$ hidden neurons, hidden weights $w \in \mathbb{R}^{H \times D}$, and $O \in \{1, 2\}$ readout weights $v \in \mathbb{R}^{O \times H}$. The network function is defined as

$$f_{w,v}(\vec{x}) = \sum_{h=1}^{H} \vec{v}_h (\langle \vec{w}_h, \vec{x} \rangle)_+, \tag{3}$$

where $(\cdot)_+$ is the ReLU nonlinearity. Importantly, the ReLU nonlinearity is homogeneous and, as a result, the network function is invariant to rescaling the hidden weights by $\alpha > 0$ and the readout weights by $\frac{1}{\alpha}$. It will therefore be useful to consider a re-parameterization in terms of the

$$\text{magnitude } m_h := v_h \|\vec{w}_h\|_2 \text{ and direction } \vec{\theta}_h := \vec{w}_h / \|\vec{w}_h\|_2. \tag{4}$$

Under this re-parameterization, an equivalent definition of the network function is given by

$$f_{m,\theta}(\vec{x}) = \sum_{h=1}^{H} m_h (\langle \vec{\theta}_h, \vec{x} \rangle)_+ = f_{w,v}(\vec{x}). \tag{5}$$

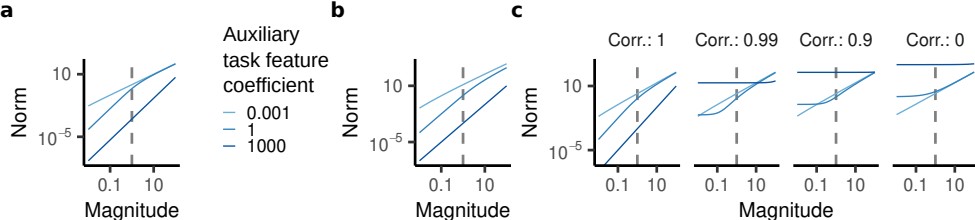

Figure 1: Theoretically derived regularization penalties. **a**, Explicit regularization penalty associated with multi-task learning. **b**, Implicit regularization penalty associated with finetuning in diagonal linear networks. **c**, Explicit regularization penalty associated with finetuning in ReLU networks. This penalty also depends on the changes in feature direction over finetuning (measured by the correlation between the unit-normalized feature directions pre vs. post finetuning).

**Training paradigms.** We consider two datasets: an auxiliary task $X^{aux} \in \mathbb{R}^{n_{aux} \times D}, \vec{y}^{aux} \in \mathbb{R}^{n_{aux}}$ and a main task $X^{main} \in \mathbb{R}^{n_{main} \times D}, \vec{y}^{main} \in \mathbb{R}^{n_{main}}$.

First, we consider **multi-task learning** (MTL): simultaneous training on both the auxiliary and main task. In this case, we consider networks with $O = 2$ outputs. The first output corresponds to the auxiliary task and the second output to the main task. Accordingly, we denote $\beta^{aux} := \beta_1$ and $\beta^{main} := \beta_2$ in the diagonal linear network and $\vec{v}^{aux} := \vec{v}_1$ and $\vec{v}^{main} := \vec{v}_2$ in the ReLU network.

Second, we consider **pretraining** on the auxiliary task and subsequent **finetuning** on the main task (PT+FT). In this case, we consider networks with a single output ($O = 1$), but re-initialize the readout weights before finetuning. Accordingly, we denote the parameters learned after pretraining by $w^{aux}$ and $v^{aux}$, and the parameters learned after finetuning by $w^{main}$ and $v^{main}$. We define $\beta^{main}, \beta^{aux}$ (for the diagonal network) and $m^{main}, \theta^{main}, m^{aux}, \theta^{aux}$ (for the ReLU network) analogously.

## 3.2 Explicit regularization penalties in multi-task learning

To theoretically understand the inductive bias of diagonal linear and ReLU networks trained with MTL, we consider the effect of minimizing the $\ell_2$ parameter norm as an approximation of the implicit bias of training with gradient descent from small initialization. We argue that this is a reasonable heuristic. First, the analogous result holds in the single-output case for infinitesimally small initialization and two layers (though not for deeper networks, see [38]). Second, for cross-entropy loss it has been shown that gradient flow on all positively homogeneous networks (including diagonal linear networks and ReLU networks) converges to a KKT point of a max-margin/min-parameter-norm objective [12]. Finally, explicit $\ell_2$ parameter norm regularization ("weight decay") is commonly used in practice, making its inductive bias important to understand in its own right as well.

We now derive the norms minimized by explicit regularization in MTL:

**Corollary 1.** *For the multi-output **diagonal linear network** defined in Eq. 2, a solution $\beta^*$ with minimal parameter norm $\|w\|_2^2 + \|v\|_2^2$ subject to the constraint that it fits the training data ($X^{main}\vec{\beta}^{main} = \vec{y}^{main}, X^{aux}\vec{\beta}^{aux} = \vec{y}^{aux}$) also minimizes the following:*

$$\beta^* = \arg\min_\beta \left( 2\sum_{d=1}^{D} \sqrt{(\beta_d^{aux})^2 + (\beta_d^{main})^2} \right) \quad s.t. \quad X^{main}\vec{\beta}^{main} = \vec{y}^{main}, \ X^{aux}\vec{\beta}^{aux} = \vec{y}^{aux}.$$

This norm is known as the group lasso [47] and denoted as $\| \cdot \|_{1,2}$. For **ReLU networks**, by an argument analogous to the one above, parameter norm minimization translates to minimizing $\sum_{h=1}^{H} \sqrt{(m_h^{aux})^2 + (m_h^{main})^2}$ (see Appendix A.1 and [44, 46]).

The norm $\| \cdot \|_{1,2}$ is plotted in Fig. 1a. We analyze its impact in Section 4.

## 3.3 Regularization penalties in finetuning

**Diagonal linear networks.** We now consider the behavior of PT+FT in overparameterized diagonal linear networks trained to minimize mean-squared error using gradient flow. We assume that the

network is initialized prior to pre-training with infinitesimal weights, and that during pretraining, network weights are optimized to convergence on the training dataset $(X^{aux}, \vec{y}^{aux})$ from the auxiliary task. After pretraining, the second-layer weights ($v_+$ and $v_-$) are reinitialized with constant magnitude $\gamma$. Further, to ensure that the network output pre-finetuning is zero (as in [38]), we set the values of corresponding positive and negative pathway weights to be equal to their sum following pretraining[1]. The network weights are further optimized to convergence on the main task dataset $(X^{main}, \vec{y}^{main})$. The dynamics of the pretraining and finetuning steps can be derived as a corollary of [38, 40]:

**Corollary 2.** *If the gradient flow solution $\vec{\beta}^{aux}$ for the diagonal linear model in Eq. 1 during pretraining fits the auxiliary task training data with zero error (i.e. $X^{aux}\vec{\beta}^{aux} = \vec{y}^{aux}$), and following reinitialization of the second-layer weights and finetuning, the gradient flow solution $\vec{\beta}^*$ after finetuning fits the main task data with zero training error (i.e. $X^{main}\vec{\beta}^{main} = \vec{y}^{main}$), then*

$$\vec{\beta}^* = \arg\min_{\vec{\beta}^{main}} \|\vec{\beta}^{main}\|_Q \quad s.t. \quad X\vec{\beta} = \vec{y},$$

$$\|\vec{\beta}^{main}\|_Q := \sum_{d=1}^{D} \left(|\beta_d^{aux}| + \gamma^2\right) q\left(\frac{2\beta_d^{main}}{|\beta_d^{aux}| + \gamma^2}\right), \quad q(z) = 2 - \sqrt{4 + z^2} + z \cdot \operatorname{arcsinh}(z/2).$$

This corollary is proven in Appendix A.2. The norm is plotted in Fig. 1b.

**ReLU networks.** We assume that after pretraining, the readout layer is re-initialized with arbitrary new weights $\vec{\gamma} \in \mathbb{R}^H$. We then characterize the solution to the finetuning task that minimizes the $\ell_2$ norm of weight changes from this initialization. This is similar to the explicit weight regularization considered in the previous section, except we now penalize weight changes from a particular initialization rather than the origin. We chose to consider this regularization penalty for two reasons. First, it is sometimes studied in the continual learning setting [30, 48]. Second, infinitesimal explicit regularization is equivalent to the implicit regularization induced by gradient descent in the case of shallow linear models [10]. While this is not true more generally, it is a useful heuristic to motivate our theoretical analysis (which we then validate using our experiments in Section 4).

We show that this finetuning solution implicitly minimizes the following penalty:

**Proposition 3.** *Consider a single-output ReLU network (see Eq. 3) with first-layer weights $\vec{w}_h^{aux} \in \mathbb{R}^D$ after pretraining, and second-layer weights re-initialized to $\vec{\gamma} \in \mathbb{R}^H$. The solution to the finetuning task that minimizes the $\ell_2$ norm of changes in the weights, i.e. minimizes $\sum_{h=1}^{H} \|\vec{w}_h - \vec{w}_h^{aux}\|_2^2 + (v_h - \gamma_h)^2$, is equivalent to the solution that minimizes*

$$R(\theta^{main}, m^{main}|\theta^{aux}, m^{aux}) := \sum_{h=1}^{H} r(\vec{\theta}_h^{main}, m_h^{main}|\vec{\theta}_h^{aux}, m_h^{aux}),$$

$$r(\vec{\theta}_h^{main}, m_h^{main}|\vec{\theta}_h^{aux}, m_h^{aux}) := (m_h^{main}/u^* - \gamma_h)^2 + (u^*)^2 + m_h^{aux} - 2u^*\sqrt{m_h^{aux}}\langle\vec{\theta}_h^{main}, \vec{\theta}_h^{aux}\rangle,$$

*where $u^*$ is the unique positive real root of*

$$-(m_h^{main})^2 + \gamma_h m_h^{main} u - m_h^{aux}\langle\vec{\theta}_h^{main}, \vec{\theta}_h^{aux}\rangle u^3 + u^4 = 0. \tag{6}$$

We prove the proposition in Appendix A.3. It implies that the regularization penalty associated with finetuning in the ReLU network only depends on the correlation $\rho_h := \langle\vec{\theta}_h^{main}, \vec{\theta}_h^{aux}\rangle$ between the first-layer feature weights before and after finetuning, and the magnitudes of the weights of these features. We plot this penalty in Fig. 1c.

## 4 Implications of the theory: multiple regimes of feature reuse

### 4.1 Sample efficiency in teacher-student models

To validate these theoretical characterizations and illustrate their consequences, we now perform experiments in a teacher-student setup. In the diagonal linear network case, we consider a linear

---

[1]Note that the need for this procedure is an idiosyncrasy of the diagonal linear network setup, and further is unnecessary if $\gamma = 0$. Following pretraining, for each input dimension, either the positive or negative pathway weights will be zero, so setting both pathway parameters to equal the sum across pathways has the effect of copying the nonzero value over to the zeroed-out pathway

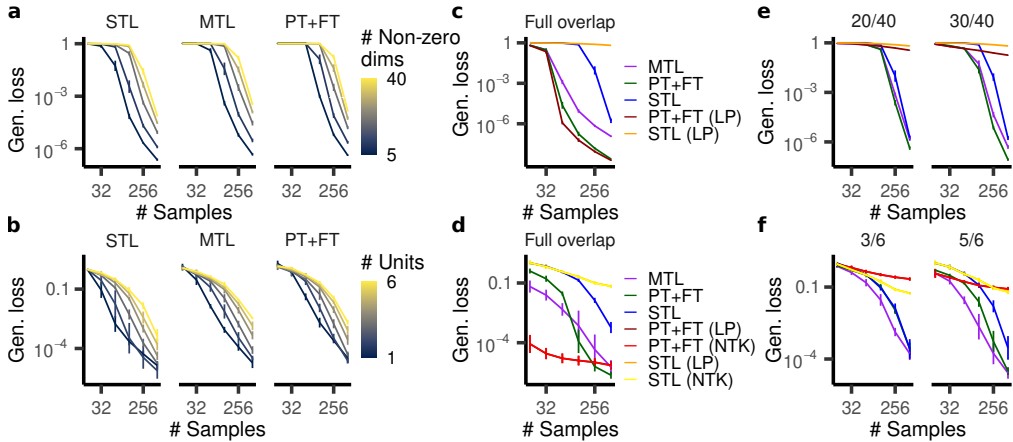

Figure 2: PT+FT and MTL benefit from feature sparsity and reuse. **a,b,** Generalization loss for a) diagonal linear networks and b) ReLU networks trained on a) a linear model with distinct active dimensions and b) a teacher network with distinct units between auxiliary and main task (STL: single-task learning). MTL and PT+FT benefit from a sparser teacher on the main task. **c,d,** Generalization loss for c) diagonal linear networks and d) ReLU networks trained on a teacher model sharing all features between the auxiliary and main task. PT+FT and MTL both generalize better than STL. **e,f,** Generalization loss for e) diagonal linear networks and f) ReLU networks trained on a teacher model with overlapping features. Networks benefit from feature sharing *and* can learn new features.

regression task defined by $\vec{w} \in \mathbb{R}^{1000}$ with a sparse set of $k$ non-zero entries. We sample two such vectors, corresponding to "auxiliary" and "main" tasks, varying the number of non-zero entries $k_{aux}$ and $k_{main}$, and the number of shared features (overlapping non-zero entries). We then uniformly sample input vectors $\vec{x} \in \mathbb{R}^{1000}$ from the unit sphere, using the ground-truth weights to generate the target. We train on 1024 auxiliary samples and vary the number of main task samples.

In the ReLU network case, we consider a "teacher" ReLU network with a sparse number of units (i.e. a low-dimensional hidden layer) and different kinds of overlaps (e.g. shared, correlated, or orthogonal features) between the auxiliary and main task. We randomly sample input data $\vec{x} \in \mathbb{R}^{15}$ from the unit sphere and use the teacher network to generate the target. We train on 1024 auxiliary samples and vary the number of main task samples. During finetuning, we randomly re-initialize the readout weights using a normal distribution with a variance of $10^{-3}\sqrt{2/H}$.

## 4.2 PT+FT and MTL benefit from sparse and shared features

**Feature sparsity.** To understand the impact of the derived penalties in the case of features either used or not used during pretraining, we consider their limit behavior for very large or very small pretrained features. First, we consider the limit $\frac{|\beta_d|}{|\beta_d^{aux}| + \gamma^2} \to \infty$ (capturing the case of a feature not used during pretraining). In this limit, the MTL penalty converges to $2|\beta_d|$. Similarly, for finetuning in diagonal linear networks, the penalty converges to $c|\beta_d|$ where $c \sim \mathcal{O}\left(\log\left(1/(|\beta_d^{aux}| + \gamma^2)\right)\right)$ (per an analysis in [38]). For new features, both networks therefore have an $\ell_1$ norm minimization bias, suggesting that they tend to learn a sparse set of new features (just like in the single-task case).

To test this insight, we consider a teacher-student setup without any shared features between the auxiliary and main task, and vary the number of main task features. Indeed, we found that both MTL and PT+FT have a more rapidly decreasing generalization loss for fewer features (just like single-task learning (STL)) (Fig. 2a). We further confirmed that they learned a sparse set of weights (Fig. 7).

The same phenomenon holds true for the ReLU network penalties as well. We considered a teacher-student network with six auxiliary task features and one to six uncorrelated main task features. Again, both MTL and PT+FT have a lower generalization loss for fewer features (Fig. 2b).

**Feature sharing.** Next, we consider the opposite limit, i.e. large pretrained features: $\frac{|\beta_d|}{|\beta_d^{aux}| + \gamma^2} \to 0$.

In this limit, both the MTL and the PT+FT penalty for diagonal linear networks converges to $\frac{\beta_d^2}{|\beta_d^{aux}|}$,

a weighted $\ell_2$ bias. This implies that the networks preferentially use large pretrained features. To test this, we consider a teacher-student setup with fully overlapping dimensions. Indeed, both MTL and PT+FT outperform STL in this case (Fig. 2c). Notably, they perform similarly to a network where we only finetune the second layer (PT+FT (Linear Probing, LP)), which exactly implements the weighted $\ell_2$ bias.

In the case of ReLU networks, we considered a teacher network with the same six features for the auxiliary and main task. Again, we found that MTL and PT+FT outperformed STL, though training a linear readout from a model with fixed features (either by using the hidden layer (PT+FT (LP)) or the neural tangent kernel (PT+FT (NTK))) generalized even better (Fig. 2d).

**Simultaneous sparsity and feature sharing.** Finally, our analysis above considered the limit behavior of each feature separately. This suggests that models should be able to 1) preferentially rely on pretrained features and 2) when necessary, learn a sparse set of new features. To test this insight, we consider partially overlapping teacher models. In diagonal linear networks, we consider teacher models with forty auxiliary and main task features, with twenty or thirty of those features overlapping (Fig. 2e). On the one hand, both PT+FT and MTL outperformed STL, indicating that they were able to benefit from the pretrained features. On the other hand, they also performed better than the PT+FT (LP) model which only finetuned the second layer (and therefore did not have a sparse inductive bias), indicating that they tended to learn a sparse set of new features. In ReLU networks, we consider teacher models with six auxiliary and main task features, varying their overlap. Again, we find that MTL and PT+FT outperformed both STL and PT+FT (LP), indicating that they benefitted from feature learning by implementing an inductive bias towards sparse and shared features.

**Differences between the MTL and PT+FT norms.** So far, we have highlighted several similarities between the MTL and PT+FT norms in diagonal linear networks: they tend towards the $\ell_1$ norm for small auxiliary features and a weighted $\ell_2$ norm for large auxiliary features. In the next section, we will highlight an important difference that arises in the intermediate regime. Here we briefly highlight a difference in the limit behavior for diagonal linear networks: the relative weights of the $\ell_1$- and weighted $\ell_2$-penalty are different between MTL and PT+FT. In particular, in the $\ell_1$ penalty limit, there is an extra factor of order $\mathcal{O}\left(\log\left(1/(|\beta_d^{aux}| + \gamma^2)\right)\right)$ in the PT+FT penalty. Assuming small initializations, this factor tends to be larger than 2, the corresponding coefficient in the MTL penalty. Thus, PT+FT is more strongly biased toward reusing features from the auxiliary task compared to MTL. We are careful to note, however, that in the case of ReLU networks this effect is complicated by a qualitatively different phenomenon with effects in the reverse direction (see Section 4.5).

## 4.3 A conservation law

We now turn our attention to the intermediate regime where the coefficients of a feature are of similar magnitude in the auxiliary and main tasks. We define two functions for a given penalty $P(\beta_d^{main}, \beta_d^{aux})$ (where $\beta_d^{main}$ is the main task feature coefficient and $\beta_d^{aux}$ is the auxiliary task feature coefficient): 1) the "$\ell$-order," $\frac{\partial \log P}{\partial \log \beta_d^{main}}$, which measures, locally, how strongly $P$ changes with increasing $\beta_d^{main}$ and 2) the "feature dependence" (FD), $\frac{\partial \log P}{\partial \log \beta_d^{aux}}$, which measures, locally, how much $P$ decreases for a larger auxiliary feature. In the previous section, we found that for $\beta_d^{aux} \to 0$, $\ell$-order $\to 1$ and FD $\to 0$, i.e. the penalty becomes $\ell_1$-like and does not depend on the magnitude of the auxiliary feature. In contrast, for $\beta_d^{aux} \to 1$, $\ell$-order $\to 2$ and FD $\to -1$, i.e. the penalty becomes $\ell_2$-like and depends inversely on the auxiliary task feature coefficient magnitude.

How are $\ell$-order and FD related for intermediate values of $\beta_d^{aux}$? Remarkably, we find an exact analytical relationship between these measures that holds for all penalties considered here:

**Proposition 4.** *For the MTL and PT+FT penalties derived in both the diagonal linear and ReLU cases, $\ell$-order $+ FD = 1$.*

See Appendix A.4 for proof.

Thus, there is an exact tradeoff between the sparsity and feature dependence even for intermediate auxiliary task feature coefficient values: an increase in sparsity (i.e. a smaller $\ell$-order) yields a corresponding decrease in feature dependence (i.e. FD becomes closer to zero).

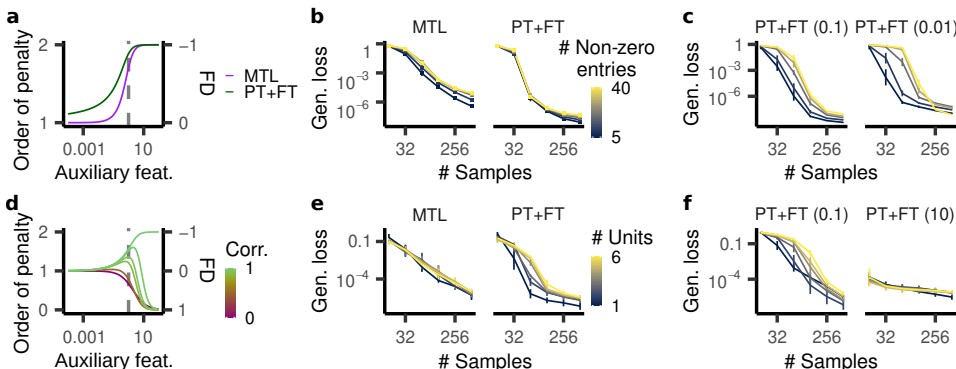

Figure 3: PT+FT (much moreso than MTL) exhibits a nested feature selection regime. **a-c**, Diagonal linear networks. **a**, $\ell$-order/feature dependence plotted for $\beta_d^{main} = 1$ and varying the auxiliary task feature coefficient. **b**, Generalization loss for models trained on a teacher with 40 active units during the auxiliary task and a subset of those units active during the main task. **c**, Generalization loss for PT+FT models whose weights are rescaled by the factor in the parentheses before finetuning. **d-f**, ReLU networks. **d**, $\ell$-order/feature dependence plotted for the explicit finetuning and MTL penalties, for $m = 1$ and varying the auxiliary task feature coefficient. **e**, Generalization loss for models trained on a teacher network with six active units on the auxiliary task and a subset of those units on the main task. **f**, Generalization loss for PT+FT models whose weights are rescaled before finetuning.

## 4.4 The nested feature selection regime

Proposition 4 predicts the existence of a novel "nested feature selection" regime: for intermediate magnitudes of the auxiliary features, the penalties should encourage both sparsity and feature dependence. To test this prediction in diagonal linear networks, we use a teacher-student setting in which all of the main task features are a subset of the auxiliary task features, i.e. $k_{main} \leq k_{aux}$, and the number of overlapping units is equal to $k_{main}$. Solving this task most efficiently involves "nested feature selection," a bias towards feature reuse *and* towards sparsity among the reused features. We plot $\ell$-order and FD for $\beta_d^{main} = 1$ and varying the auxiliary task feature coefficient (Fig. 3a) and find that for $\beta_d^{aux} \approx 1$, both norms exhibit "lazy regime"-like behavior ($\ell$-order of around 2, and FD near 0). This predicts that neither MTL nor PT+FT networks should be able to benefit from nested sparsity task structure in this regime, which we confirm empirically: as $k_{main}$ decreases, the networks' sample efficiency does not become substantially better (Fig. 3b).

However, for features with auxiliary task coefficients that are moderately smaller than their main task coefficients, Fig. 3a suggests a broad regime where the $\ell$-order is closer to 1 (incentivizing sparsity), but feature dependence remains high. We can produce this behavior in these tasks by rescaling the weights of the network following pretraining by a factor less than 1. In line with the prediction of the theory, performing this manipulation enables PT+FT to leverage sparse structure *within* auxiliary task features (Fig. 3c), even while retaining their ability to privilege features learned during pretraining above others (see Fig. 9). By contrast, this regime is much narrower for MTL (Fig. 3a).

For ReLU networks, the MTL penalty is the same as in diagonal linear networks. We plot the regularization penalty derived for PT+FT, conditioned on various correlations $\langle \theta_h^{main}, \theta_h^{aux} \rangle$ between the post-pretraining and post-finetuning feature weights (Fig. 3d). For features that are fully aligned before and after pretraining, this penalty is again $\ell_2$-like for $m^{aux} \approx 1$. However, in most cases, these features change direction at least a little bit, and we find that in that case the penalty is more $\ell_1$-like while still remaining sufficiently feature-dependent. This suggests that a nested feature selection regime may arise in ReLU networks even when auxiliary task feature coefficients have comparable magnitude to main task feature coefficients. To test this insight, we considered a set of tasks in which the main task solution relies exclusively on a subset of auxiliary task features. We found that PT+FT (even without any weight rescaling) was able to benefit from the nested sparsity structure, but MTL was not (Fig. 3e).[2] Performing weight rescaling in either direction following pretraining uncovers the

---

[2]We note that we had observed this behavior before deriving the PT+FT regularization penalty for ReLU networks. Unlike all other described experiments (for which we derived the described predictions from inspecting the norms before running any simulations), this is therefore a postdiction rather than a prediction.

initialization-insensitive (FD near 0), sparsity-biased ($\ell$-order near 1) rich / feature-learning regime and the initialization-biased (FD near $-1$), no-sparsity-bias ($\ell$-order near 2) lazy learning regime (Fig. 3f). This suggests that for different architectures and different tasks, different rescaling values may be required to enter the nested feature selection regime.

### 4.5   PT+FT, but not MTL, in ReLU networks benefits from correlated features

The regularization associated with PT+FT yields benefits even when main/auxiliary task directions are correlated but not identical (Fig. 1c). In contrast, MTL cannot softly share features as it encodes correlated features with entirely distinct units. To test this hypothesis, we conduct experiments in which the ground-truth auxiliary and main tasks rely on correlated but distinct features. Indeed, we find PT+FT outperforms STL in this case, whereas MTL only does so in the low-sample setting (Fig. 4a). Notably, if features are identical, MTL outperforms PT+FT in the low-sample setting (Fig. 2d). Thus, PT+FT (compared to MTL) trades off the flexibility to "softly" share features for reduced sample-efficiency when such flexibility is not needed.

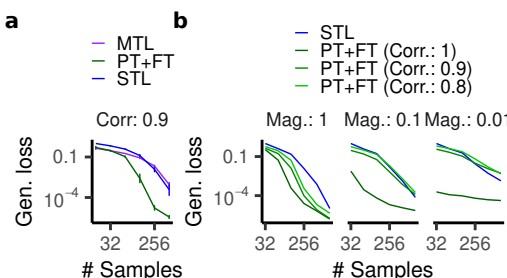

Figure 4: PT+FT, but not MTL, in ReLU networks benefits from correlated features. **a**, Generalization loss for main task features that are correlated (0.9 cosine similarity) with the auxiliary task features. PT+FT outperforms both MTL and STL. **b**, Generalization loss for main task features with varying correlation and magnitude (mag.). PT+FT only outperforms STL if the features are either identical in direction or identical in magnitude.

The analysis in Fig. 3d would predict that ReLU networks can no longer benefit from correlated features if the magnitude of auxiliary task features is much higher than that of main task features. To test this, we varied the magnitude of the main task features and their correlation with auxiliary task features (Fig. 4b). We found that for lower magnitudes, PT+FT still improved performance for the case of identical but not correlated feature directions, confirming our prediction.

## 5   Weight rescaling in deep networks gives rise to nested feature selection

Our analysis has focused on shallow networks trained on synthetic tasks. To test the applicability of our insights, we conduct experiments with convolutional networks (ResNet-18, [49]) on a vision task (CIFAR-100, [50]), using classification of two image categories (randomly sampled for each training run) as the primary task and classification of the other 98 as the auxiliary task. As in our experiments above, MTL and PT+FT improve sample efficiency compared to single-task learning (Fig. 5a).

Our findings in Section 4.4 indicate that the nested feature selection bias of PT+FT can be exposed or masked by rescaling the network weights following pretraining. Such a bias may be beneficial when the main task depends on a small subset of features learned during pretraining, as may often be the case in practice. We experiment with rescaling in our CIFAR setup. We find that rescaling values less than 1 improve finetuning performance (Fig. 5b). These results suggest that rescaling network weights before finetuning may be practically useful. We corroborate this hypothesis with additional experiments using networks pre-trained on ImageNet [51] (see Fig. 10).

To facilitate comparison of the phenomenology in deep networks with our teacher-student experiments above, we propose a signature of nested feature selection that can be characterized without knowledge of the underlying feature space (since the correct feature basis to analyze is less clear in multi-hidden-layer entworks). Specifically, we propose to measure (1) the *dimensionality* of the network representation pre- and post-finetuning, and (2) the extent to which the representational structure post-finetuning is shared with / inherited from that of the network following pretraining prior to finetuning. We employ the commonly used *participation ratio* (PR, [52]) as a measure of dimensionality, and the *effective number of shared dimensions* (ENSD, [53]) as a soft measure of the number of aligned principal components between two representations. Intuitively, the PR and ENSD of network representations pre- and post-finetuning capture the key phenomena of the nested feature selection regime: we expect the dimensionality of network after finetuning to be

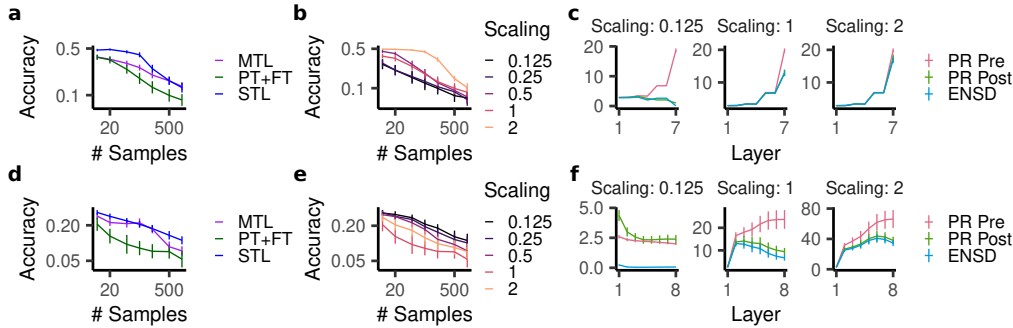

Figure 5: Experiments in deep neural networks trained on CIFAR-100: **a-c**, ResNet-18, **d-f**, ViT. **a**,**d**, Accuracy for MTL, PT+FT, and STL in a) ResNet-18 and d) ViT. **b**,**e** Accuracy for PT+FT with weight rescaling in b) ResNet-18 and e) ViT. **c**,**f** The participation ration of c) ResNet-18's and f) ViT's layers before and after finetuning (PR Pre and PR Post) as well as their ENSD.

lower than after pretraining ($PR(\mathbf{X}_{FT}) < PR(\mathbf{X}_{PT})$), and for nearly all of the representational dimensions expressed by the network post-finetuning to be inherited from the network state after pretraining ($ENSD(\mathbf{X}_{PT}, \mathbf{X}_{FT}) \approx PR(\mathbf{X}_{FT})$). We validate that this description holds in our nonlinear teacher-student experiments with networks in the nested feature selection regime (Fig. 11).

Remarkably, we find that the ResNet-18 exhibits the same phenomenology, but only for weights rescaled by a small value (Fig. 5c). This supports the hypothesis that the observed benefits of rescaling indeed arise from pushing the network into the nested feature selection regime.

Finally, we conduct the same experiment for Vision Transformers (ViT) [54]. We confirm that PT+FT improves performance over single-task learning, though MTL offers no similar benefit in this case (Fig. 5d). We further find that rescaling before finetuning (both by larger and smaller values) decreases generalization performance (Fig. 5e). Notably, our representational analysis reveals that rescaling by a larger value yields a higher-dimensional subspace both before and after finetuning, whereas rescaling by a smaller value yields a lower-dimensional subspace, but pushes the effective number of shared dimensions down substantially (Fig. 5f). This indicates that a rescaling value of 1 may already give rise to the nested feature selection regime and rescaling by a smaller value pushes the ViT towards the pure feature learning regime. Taken together, our results suggest finetuning performance is best when networks operate in the nested feature selection regime, and weight rescaling can push networks into this regime when it does not arise naturally.

## 6 Conclusion

In this work we have provided a detailed characterization of the inductive biases associated with two common training strategies, MTL and PT+FT, in diagonal linear and ReLU networks. These biases incentivize both feature sharing and sparse task-specific feature learning. In the case of PT+FT, we characterized a novel *nested feature selection* learning regime which encourages sparsity *within* features inherited from pretraining. This insight motivates a simple technique for improving PT+FT performance by pushing networks into this regime, which shows promising empirical results.

There are several avenues for extending our theoretical work: for example, connecting our derived penalties for ReLU networks (which assumed explicit parameterization) to the implicit regularization induced by dynamics of gradient descent, and extending our theory to the case of cross-entropy loss. In addition, more work is needed to extend our theory to more complex tasks and larger models. For instance, we are interested in investigating how the weight magnitudes required to enter the nested feature regime depend on architecture and properties of the tasks — we observed a difference between the rescaling values required for ResNets and Vision Transformers, which our shallow network theory is unable to speak to. Better understanding the conditions needed for nested feature selection could also inspire more sophisticated interventions than our weight rescaling trick. Finally, we considered a particularly simple multi-task setup, with identical formats between auxiliary and main tasks — our theory could be extended to cases where the different tasks use different objectives. Nevertheless, our work already provides new and practical insights into multi-task learning and finetuning.

## Acknowledgments

We are grateful to the members of the Center for Theoretical Neuroscience for helpful comments and discussions. The work was supported by NSF 1707398 (Neuronex), Gatsby Charitable Foundation GAT3708, and the NSF AI Institute for Artificial and Natural Intelligence (ARNI).

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

## A Derivation of regularization penalties

### A.1 Multi-task learning in ReLU networks

Multi-task ReLU networks with a shared feature layer and $O$ outputs can be written as

$$f_w(x) = \sum_{h=1}^{H} \vec{v}_h(\langle w_h, \vec{x}\rangle)_+ = \sum_{h=1}^{\vec{H}} \vec{m}_h(\langle \vec{\theta}_h, \vec{x}\rangle)_+, \tag{7}$$

$$\vec{m}_h = \vec{v}_h \|\vec{w}_h\|_2, \vec{\theta}_h = \vec{w}_h / \|\vec{w}_h\|_2, \tag{8}$$

where $\vec{w}_h, m_h \in \mathbb{R}^O$. We are interested in how minimizing the parameter norm $\sum_{h=1}^{H} \|\vec{w}_h\|_2^2 + \|\vec{v}_h\|_2^2$ maps to minimizing a norm over the solution weights $\vec{m}$. For a given $\vec{m}_h \in \mathbb{R}^O$, the norm of the input weight $\|\vec{w}_h\|_2$ is a free parameter $z_h > 0$, as we can set $\vec{v}_h := \vec{m}_h / z$. As any value for $z_h$ leads to the same function, we choose $z_h$ so as to minimize

$$\|\vec{w}_h\|_2^2 + \|\vec{v}_h\|_2^2 = z_h^2 + \|\vec{m}_h\|_2^2 / z_h^2. \tag{9}$$

Setting the derivative to zero implies

$$z_h^2 = \|\vec{m}_h\|_2. \tag{10}$$

As a result,

$$\sum_{h=1}^{H} \|\vec{w}_h\|_2^2 + \|\vec{v}_h\|_2^2 = 2\sum_{h=1}^{H} \|\vec{m}_h\|_2, \tag{11}$$

where the right-hand side of the equation is the $\ell_{1,2}$ norm.

### A.2 Finetuning in diagonal linear networks

**Corollary 2.** *If the gradient flow solution $\vec{\beta}^{aux}$ for the diagonal linear model in Eq. 1 during pretraining fits the auxiliary task training data with zero error (i.e. $X^{aux}\vec{\beta}^{aux} = \vec{y}^{aux}$), and following reinitialization of the second-layer weights and finetuning, the gradient flow solution $\vec{\beta}^*$ after finetuning fits the main task data with zero training error (i.e. $X^{main}\vec{\beta}^{main} = \vec{y}^{main}$), then*

$$\vec{\beta}^* = \underset{\vec{\beta}^{main}}{\arg\min} \|\vec{\beta}^{main}\|_Q \quad s.t. \quad X\vec{\beta} = \vec{y},$$

$$\|\vec{\beta}^{main}\|_Q := \sum_{d=1}^{D} \left(|\beta_d^{aux}| + \gamma^2\right) q\left(\frac{2\beta_d^{main}}{|\beta_d^{aux}| + \gamma^2}\right), \quad q(z) = 2 - \sqrt{4 + z^2} + z \cdot \text{arcsinh}(z/2).$$

*Proof.* We derive this result in two steps: first, we note that after pretraining (because of the infinitesimal initial weight magnitudes), one of the pathways remains at zero and one pathway encodes the effective linear predictor $\beta^{aux}$. Thus, the first hidden layer has the weights $\sqrt{\beta^{aux}}$, which yields the new weight magnitude along the positive and negative pathway (due to the summing operation we perform following finetuning, which ensures that the weights of both pathways are set to $\sqrt{\beta^{aux}}$). Having set the readout initialization to $\gamma$, we then apply Theorem 4.1 in [40]. $\square$

### A.3 Finetuning in ReLU networks

**Proposition 3.** *Consider a single-output ReLU network (see Eq. 3) with first-layer weights $\vec{w}_h^{aux} \in \mathbb{R}^D$ after pretraining, and second-layer weights re-initialized to $\vec{\gamma} \in \mathbb{R}^H$. The solution to the finetuning task that minimizes the $\ell_2$ norm of changes in the weights, i.e. minimizes $\sum_{h=1}^{H} \|\vec{w}_h - \vec{w}_h^{aux}\|_2^2 + (v_h - \gamma_h)^2$, is equivalent to the solution that minimizes*

$$R(\theta^{main}, m^{main} | \theta^{aux}, m^{aux}) := \sum_{h=1}^{H} r(\vec{\theta}_h^{main}, m_h^{main} | \vec{\theta}_h^{aux}, m_h^{aux}),$$

$$r(\vec{\theta}_h^{main}, m_h^{main} | \vec{\theta}_h^{aux}, m_h^{aux}) := (m_h^{main}/u^* - \gamma_h)^2 + (u^*)^2 + m_h^{aux} - 2u^* \sqrt{m_h^{aux}} \langle \vec{\theta}_h^{main}, \vec{\theta}_h^{aux} \rangle,$$

*where $u^*$ is the unique positive real root of*

$$-(m_h^{main})^2 + \gamma_h m_h^{main} u - m_h^{aux} \langle \vec{\theta}_h^{main}, \vec{\theta}_h^{aux} \rangle u^3 + u^4 = 0. \tag{6}$$

*Proof.* We define the norm of the deviation from the parameters at initialization as

$$R(w, v | w^{aux}, \gamma) := \sum_{h=1}^{H} (v_h - \gamma)^2 + \|\vec{w}_h - w_h^{aux}\|_2^2. \tag{12}$$

Note that $w$ and $v$ are parameterized in a redundant manner, as we can multiply $v_h$ by a constant $a > 0$ and obtain the same function as long as we divide $\vec{w}_h$ by the same constant. To parameterize a function in a unique fashion, we describe it in terms of the normalized hidden weight $\vec{\theta}_h := \vec{w}_h / \|\vec{w}_h\|_2$ and the (signed) magnitude $m_h := \|\vec{w}_h^{(1)}\|_2 v_h$. We wish to compute the norm of the deviation from the initialization in terms of this parameterization. Specifically, we compute

$$\tilde{R}(\vec{\theta}, m | w^{aux}, \gamma) := \min_{w,v} R(w, v | w^{aux}, \gamma) \text{ s.t. } \forall_h \vec{\theta}_h = \vec{w}_h / \|\vec{w}_h\|_2, m_h = \|\vec{w}_h\|_2 v_h. \tag{13}$$

Note that Woodworth *et al.* [38] solve a version of this optimization problem for diagonal linear networks (though only for the case in which corresponding first and second layer weights are initialized with equal magnitude).

As this optimization problem decomposes over hidden units, we can solve it for each hidden unit individually:

$$\tilde{R}(\vec{\theta}, m | w^{aux}, \gamma) = \sum_{h=1}^{H} \tilde{r}(\vec{\theta}_h, m_h | \vec{w}_h^{aux}, \gamma), \tag{14}$$

$$\tilde{r}(\vec{\theta}_h, m_h | \vec{w}_h^{aux}, \gamma) := \min_{\vec{w}_h, v_h} (v_h - \gamma)^2 + \|\vec{w}_h - \vec{w}_h^{aux}\|_2^2 \tag{15}$$

$$\text{s.t. } \vec{\theta}_h = \vec{w}_h / \|\vec{w}_h\|_2, m_h = \|\vec{w}_h\|_2 v_h. \tag{16}$$

We can express this as

$$\tilde{r}(\vec{\theta}_h, m_h | w_h^{aux}, \gamma) = \min_{u>0} p(u), \quad p(u) := (m_h/u - \gamma)^2 + \left\| u\vec{\theta}_h - \vec{w}_h^{aux} \right\|_2^2. \tag{17}$$

Note that this function is smooth and diverges to $\infty$ both for $u \to \infty$ and $u \to 0$. As long as we have a unique positive stationary point $u^*$, this stationary point must therefore be a minimum. We simplify

$$p(u) = (m_h/u - \gamma)^2 + u^2 + \|\vec{w}_h^{aux}\|_2^2 - 2u\langle \vec{\theta}_h, \vec{w}_h^{aux} \rangle, \tag{18}$$

and, defining $m_h^{aux} := \|\vec{w}_h^{aux}\|_2$, and the cosine similarity between corresponding pretraining and finetuning task weights $\rho_h := \frac{\langle \vec{w}_h^{(1)}, \vec{w}_h^{aux} \rangle}{\|\vec{w}_h^{(1)}\|_2 \|\vec{w}_h^{aux}\|_2}$, write this as

$$p(u) = (m_h/u - \gamma)^2 + u^2 + (m_h^{aux})^2 - 2u m_h^{aux} \rho_h. \tag{19}$$

We then determine the stationary points by computing the derivative

$$p'(u) = -2(m_h/u - \gamma)m_h/u^2 + 2u - 2m_h^{aux}\rho_h, \tag{20}$$

and setting it to zero. Simplifying this equation (and multiplying by $u^3/2$) yields

$$-m_h^2 + \gamma m_h u - m_h^{aux}\rho_h u^3 + u^4 = 0. \tag{21}$$

We can compute the solution to this quartic equation, e.g. using `np.roots` and select the (unique, in all empirical cases we explored) positive real root, $u^*$. We can then compute the original norm by plugging in $u^*$:

$$\tilde{r}(\vec{\theta}_h, m_h | \vec{w}_h^{aux}, \gamma) = (m_h/u^* - \gamma)^2 + u^{*2} + (m_h^{aux})^2 - 2u^* m_h^{aux} \rho_h. \tag{22}$$

$\square$

## A.4 Proof of Proposition 4

**Proposition 4.** *For the MTL and PT+FT penalties derived in both the diagonal linear and ReLU cases, $\ell$-order $+ FD = 1$.*

*Proof.* Below, for simplicity's sake, we always denote $\beta^{main} = \beta$, $m^{main} = m$, and $\theta^{main} = \theta$.

**MTL penalty.** We first consider the MTL penalty, which is identical for diagonal linear networks and ReLU networks. We assume (without loss of generalization) $m^{aux} > 0$. In that case,

$$\ell\text{-order} = \frac{\partial \log\left(\sqrt{(m^{aux})^2 + m^2} - m^{aux}\right)}{\partial \log m} = \tag{23}$$

$$\frac{m}{\sqrt{(m^{aux})^2 + m^2} - m^{aux}} \frac{m}{\sqrt{(m^{aux})^2 + m^2}} = \frac{m^2}{(m^{aux})^2 + m^2 - m^{aux}\sqrt{(m^{aux})^2 + m^2}},$$

and

$$\text{FD} = \frac{\partial \log\left(\sqrt{(m^{aux})^2 + m^2} - m^{aux}\right)}{\partial \log m^{aux}} = \tag{24}$$

$$\frac{m^{aux}}{\sqrt{(m^{aux})^2 + m^2} - m^{aux}}\left(\frac{m^{aux}}{\sqrt{(m^{aux})^2 + m^2}} - 1\right) = \frac{(m^{aux})^2 - m^{aux}\sqrt{(m^{aux})^2 + m^2}}{(m^{aux})^2 + m^2 - m^{aux}\sqrt{(m^{aux})^2 + m^2}}.$$

Thus, $\ell\text{-order} + \text{FD} = 1$.

**Finetuning in diagonal network.** We assume (without loss of generalization) $\beta_d^{aux} > 0$ and derive

$$\ell\text{-order} = \frac{\partial \log\left((|\beta_d^{aux}|)\, q\left(\frac{2\beta_d}{\beta_d^{aux}}\right)\right)}{\partial \log \beta_d} = \frac{2\beta_d}{(\beta_d^{aux})^2 q\left(\frac{2\beta_d}{|\beta_d^{aux}|}\right)} q'\left(\frac{2\beta_d}{|\beta_d^{aux}|}\right). \tag{25}$$

and

$$\text{FD} = \frac{\partial \log\left(|\beta_d^{aux}| q\left(\frac{2\beta_d}{|\beta_d^{aux}|}\right)\right)}{\partial \log |\beta_d^{aux}|} \tag{26}$$

$$\frac{1}{q\left(\frac{2\beta_d}{|\beta_d^{aux}|}\right)}\left(q\left(\frac{2\beta_d}{|\beta_d^{aux}|}\right) - q'\left(\frac{2\beta_d}{|\beta_d^{aux}|}\right)\frac{2\beta_d}{(\beta_d^{aux})^2}\right) = 1 - \frac{2\beta_d}{(\beta_d^{aux})^2 q\left(\frac{2\beta_d}{|\beta_d^{aux}|}\right)} q'\left(\frac{2\beta_d}{|\beta_d^{aux}|}\right).$$

Thus, $\ell\text{-order} + \text{FD} = 1$.

**Finetuning in ReLU networks.** We define

$$\tilde{r}(\vec{\theta}_h, m_h, u | \vec{\theta}_h^{aux}, m_h^{aux}) := (m_h/u - \gamma)^2 + u^2 + m_h^{aux} - 2u\sqrt{m_h^{aux}}\langle\vec{\theta}_h, \vec{\theta}_h^{aux}\rangle, \tag{27}$$

where, without loss of generalization, we assume that $m_h^{aux} > 0$ and note that

$$r(\vec{\theta}_h, m_h | \vec{\theta}_h^{aux}, m_h^{aux}) = \tilde{r}(\vec{\theta}_h, m_h, u^* | \vec{\theta}_h^{aux}, m_h^{aux}), \tag{28}$$

where $u^*$ is the critical point, depending on $\vec{\theta}_h, m_h, \vec{\theta}_h^{aux}, m_h^{aux}$. By the chain rule,

$$\frac{\partial r(\vec{\theta}_h, m_h | \vec{\theta}_h^{aux}, m_h^{aux})}{\partial m_h} = \frac{\partial \tilde{r}(\vec{\theta}_h, m_h, u^* | \vec{\theta}_h^{aux}, m_h^{aux})}{\partial m_h} + \frac{\partial \tilde{r}(\vec{\theta}_h, m_h, u^* | \vec{\theta}_h^{aux}, m_h^{aux})}{\partial u^*}\frac{\partial u^*}{m_h}, \tag{29}$$

and the analogous statement is true for $m_h^{aux}$.

However, by definition,

$$\frac{\partial \tilde{r}(\vec{\theta}_h, m_h, u^* | \vec{\theta}_h^{aux}, m_h^{aux})}{\partial u^*} = 0. \tag{30}$$

Thus,

$$\ell\text{-order} = \frac{m_h}{r}\frac{\partial \tilde{r}}{\partial m_h} = \frac{2m_h^2}{r(u^*)^2}, \tag{31}$$

$$\text{FD} = \frac{m_h^{aux}}{r}\frac{\partial \tilde{r}}{\partial m_h^{aux}} = \frac{m_h^{aux}}{r}\left(1 - \frac{\rho_h u^*}{\sqrt{m_h^{aux}}}\right) = \frac{1}{r}\left(m_h^{aux} - \rho_h\sqrt{m_h^{aux}}u^*\right).$$

To prove the statement, we therefore must prove that

$$\frac{2m_h^2}{(u^*)^2} + m_h^{aux} - \rho\sqrt{m_h^{aux}}u^* = r = \frac{m_h^2}{(u^*)^2} + (u^*)^2 + (m_h^{aux}) - 2u^*m_h^{aux}\rho_h. \qquad (32)$$

To do so, we leverage that (30) implies

$$-\frac{m_h^2}{(u^*)^2} + (u^*)^2 - u^*\sqrt{m_h^{aux}}\rho_h = 0, \qquad (33)$$

and therefore

$$\frac{m_h^2}{(u^*)^2} = (u^*)^2 - u^*\sqrt{m_h^{aux}}\rho_h = 0. \qquad (34)$$

Plugging this into (32) yields

$$\frac{2m_h^2}{(u^*)^2} + m_h^{aux} - \rho\sqrt{m_h^{aux}}u^* = \frac{m_h^2}{(u^*)^2} + (u^*)^2 + m_h^{aux} - 2\rho\sqrt{m_h^{aux}}u^* = r \qquad (35)$$

and completes the proof. □

# B  Detailed methods

We trained all networks with PyTorch [55].

## B.1  Diagonal linear networks

We train the diagonal linear networks until they have reached a mean squared error below $10^{-10}$, for pretraining, finetuning, multi-task learning and single-task learning. As a rough heuristic for determining the learning rate, we begin at a learning rate of $10^6$ and divide by ten whenever the loss exceeds 100 (indicating divergence).

## B.2  ReLU networks

We train ReLU networks with 1000 hidden units using the same learning rate heuristic and train until the mean squared error has reached a threshold of $10^{-8}$.

## B.3  ResNets

We consider a ResNet-18 in the standard PyTorch implementation and train it with gradient descent with a learning rate of $10^{-3}$ and momentum of 0.9.

## B.4  Vision Transformers

We consider a Vision Transformer with seven layers, eight heads, 384 hidden dimensions and an MLP with 1,536 hidden dimensions. We train the transformer using the Adam optimizer with a learning rate of $10^{-3}$ and weight decay 0.00005.

## B.5  Reproducibility

We provide a code database that enables reproducing all data used to create the main figures. We used one CPU for each diagonal network (total number of experiments: 5,544) and ReLU network experiment (total number of experiments: 6,740) and all of them took up to four hours. We used on GPU for the CIFAR and ViT experiments and all of them took up to twelve hours (total number of experiments: 5,000). As a rough (upper-bound) estimate, the experiments therefore required 50,000 hours of CPU time and 60,000 hours of GPU time.

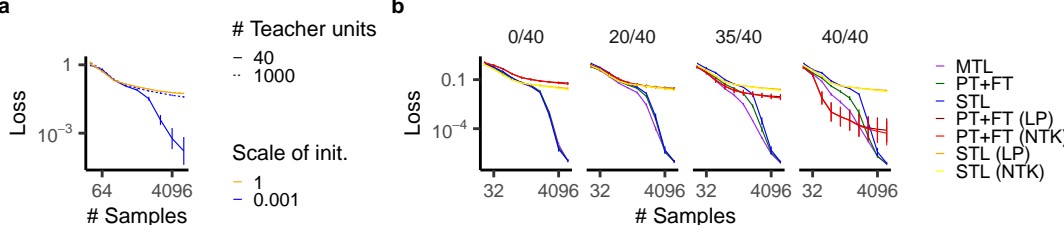

Figure 6: Larger-scale teacher-student experiments. **a**, Generalization loss of shallow ReLU networks trained on data from a ReLU teacher network. **b**, Generalization loss for different numbers of overlapping features (out of 40 total) between main and auxiliary tasks. NTK indicates the (lazy) tangent kernel solution. This is comparable to Fig. 2d, except with more teacher units and more data.

## C  Robustness of main results to choice of number of auxiliary task samples and input dimension

To increase confidence that our main results are robust to the number of data samples used (1024 auxiliary task samples and up to 1024 main task samples in most of our experiments), and the number of ground-truth units in the teacher network (6), we repeated the experiments of Fig. 2d with 8192 auxiliary task samples and 40 ground-truth features. Indeed, in this setting the rich regime also helps with generalization if and only if the teacher units are sparse (Fig. 6a). Further, MTL and PT+FT tend to outperform STL if the features are overlapping and MTL tends to outperform PT+FT (Fig. 6b). In particular, the finetuned networks still benefit from feature learning, especially if some features are novel.

## D  Analysis of learned solutions in linear and nonlinear networks

Our theory predicts inductive biases towards solutions that minimize norms, often either $\ell_1$-like (incentivizing sparsity) or $\ell_2$-like. Our experiments in the main text corroborate these description by analyzing how sample complexity depends on the feature sparsity of the ground-truth task solution, and how the sparse feature structures of the main and auxliary tasks relate. However, this evidence for sparsity biases (or lack thereof) is indirect; here we present more direct analyses of the learned solutions in linear and nonlinear networks that support the account we provide in the main text.

### D.1  Diagonal linear networks

To check whether the implicit regularization theory is a good explanation for these performance results, we directly measured the $\ell_{1,2}$ and $Q$ norms of the solutions learned by networks, compared to the corresponding penalties of the ground truth weights. In Fig. 7a we see that as the amount of training data increases, the norms all converge to that of the ground truth solution, but in the low-sample regime, MTL and PT+FT find solutions with lower values of their corresponding norm than the ground-truth function, consistent with the implicit regularization picture (by contrast, STL does not consistently find solutions with lower values of these norms than the ground truth).

Our theory predicts that weight rescaling by a factor less than 1.0 following pretraining reduces the propensity of the network to share features between auxiliary and main tasks during finetuning. We confirm that this is the case in Fig. 7b by analyzing the overlap between the auxiliary task features and the learned linear predictor for the main task.

In Fig. 7c we show that our measure of effective sparsity of learned solutions in diagonal linear networks effectively distinguishes between networks trained in the feature selection regime and networks trained with linear probing (only training second-layer weights). Moreover, in Fig. 7d we show that the L1 norm of the solution increases with the training sample size, consistent with a bias towards L1 minimization. There is an interesting discrepancy between the behavior of the sparsity of the solutions (nonmonotonic, see Fig. 7c) and the L1 norm (largely monotonic, see Fig. 7d). This is attributable to the discrepancy between the L1 norm (which diagonal linear networks in the rich regime are biased to minimize) and sparsity (for which L1 norm is only a proxy).

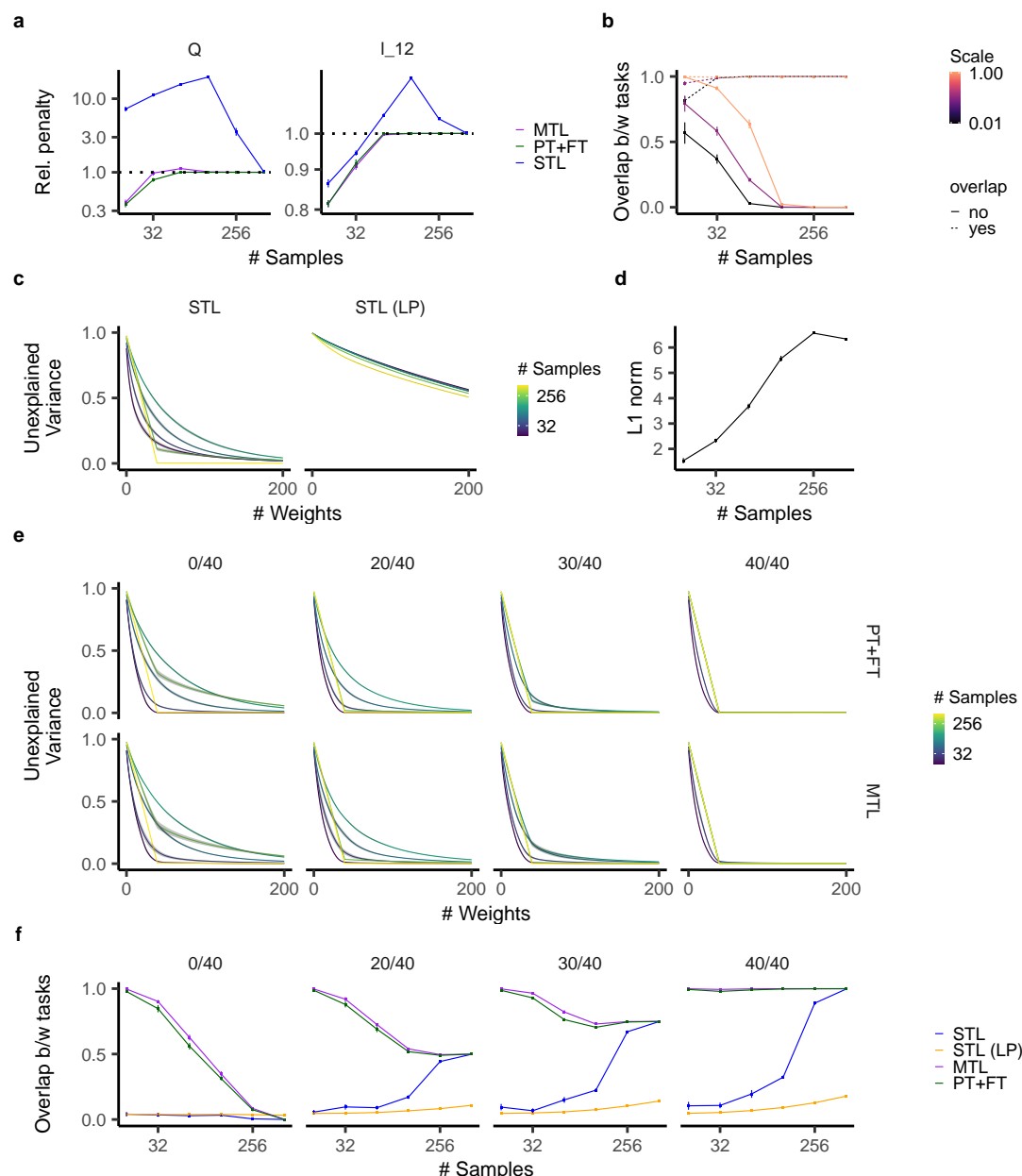

Figure 7: **a**, $\ell_{1,2}$ norm and $Q$ penalty for MTL, STL, and PT+FT networks from Fig. 2b (40/40 overlapping features case). **b**, Proportion of the weight norm in the 40 dimensions relevant for the auxiliary task, for the networks in Fig. 3b,c. Weight rescaling decreases this overlap. **c**, Proportion of variance concentrated in the top $k$ weights, as a function of $k$, for training on a single-task. When both layers are trained from small initialization (STL), this variance decreases much more rapidly than for pure linear readout training (STL (LP)), demonstrating the sparsity of the learned solution. **d**, L1 norm for STL as a function of the number of samples. **e**, Proportion of variance across different overlaps and for different learning setups (see also Fig. 2b). The rapid decrease in variance demonstrates the sparsity of the learned solutions both for PT+FT and MTL. **f**, Proportion of weight norm in the 40 dimensions relevant for the auxiliary task (see also Fig. 2b).

In Fig. 7e we show the sparsity of networks trained on teacher with different values of the number of overlapping features between main and auxiliary tasks (each of which uses 40 features). We find that learned solutions across a range of overlaps are as sparse as using single-task learning (see Fig. 7c)

when task features do not overlap (0/40 case) and more sparse otherwise (on account of the bias toward reuse of the sparse features learned during the auxiliary task, see next paragraph).

In Fig. 7f we show the overlap between features active on either task, finding that learned main task solutions are biased to share auxiliary task features when few samples are available.

## D.2 ReLU networks

We adopt a clustering-based approach to analyzing the effective sparse structure of learned task solutions. Specifically, for a given trained network, we perform k-means clustering with a predetermined value of $K$ clusters on the normalized input weights to each hidden-layer neuron in the network[3]. We measure the extent to which $K$ cluster centers are able to explain the variance in input weights across hidden units; the fraction of variance left unexplained is commonly referred to as the "inertia." For values of $K$ at which the inertia is close to zero, we can say that (to a good approximation) the network effectively makes use of at most $K$ nonlinear features.

### D.2.1 Single-task learning: rich inductive bias yields clusters of similarly tuned neurons that approximate sparse ground-truth features

In the single-task learning case, we measure the inertia of trained networks as a function of $K$. We find that for networks in the rich regime (small initialization scale), for tasks with sparse ground-truth (six units in the ReLU teacher network), the networks do indeed learn solutions that make use of approximately six nonlinear features (Fig. 8a). For tasks with many (1000) units in the teacher network, the network finds solutions that use a small number of feature clusters when main task samples are limited, but gradually uses more clusters as the number of samples is increased (Fig. 8a), at which point the network matches the teacher function very well. This bias towards sparser-than-ground-truth solutions given insufficient data corroborates our claim of an inductive bias towards sparse solutions. By contrast, networks in the lazy learning regime (large initialization scale) display no such bias, corroborating our claim that the sparse $\ell_1$-like inductive bias is a property of the rich regime but not the lazy regime. Interestingly, in the sparse ground-truth case learned solutions are relatively less sparse for an intermediate number of training examples. This may arise because an $\ell_1$-like inductive bias is not exactly the same as a bias toward sparse solutions over nonlinear features, particularly when training data is limited. We leave an in-depth investigation of this phenomenon to future work.

Our clustering analysis allows us to measure the extent to which the effective features employed by the network (cluster centers) are aligned with the ground-truth task features. Specifically, for each teacher unit, we compute an "alignment score" between teacher and student networks by taking each teacher unit, measuring its cosine similarity with all the cluster centers, choosing the maximum value, and averaging this quantity across all teacher units. We find that the learned feature clusters are indeed highly aligned with the ground-truth teacher features in the sparse ground-truth case, and moreso as the number of main task samples (and consequently task performance) increases (Fig. 8b).

### D.2.2 Pretraining+finetuning finds sparse solutions and improves alignment of feature clusters learned during pretraining

We find that pretraining+finetuning improves performance over single-task learning when main and auxiliary task features are shared (or correlated), and maintains an apparent bias toward sparsity in new task-specific features. To corroborate these claims, we performed our clustering analysis on the solutions learned through PT+FT. We find that the solutions learned are indeed quite sparse (comparable to the sparsity of solutions learned by single-task learning), even when the auxiliary task and main task features are disjoint (Fig. 8c). Moreover, we find that MTL also learns sparse solutions (Fig. 8d). In particular, as expected, the effective features on tasks with overlapping features is equal to the number of total unique features. Moreover, we observe that when main task and auxiliary task features are shared, PT+FT and MTL networks exhibit higher alignment between learned features and ground-truth features than single-task-trained networks, especially when main task samples are

---

[3]weighting the importance of each unit to the k-means objective by the weight of its contribution to the network's input-output function, specifically the magnitude of the product of its associated input and output weights

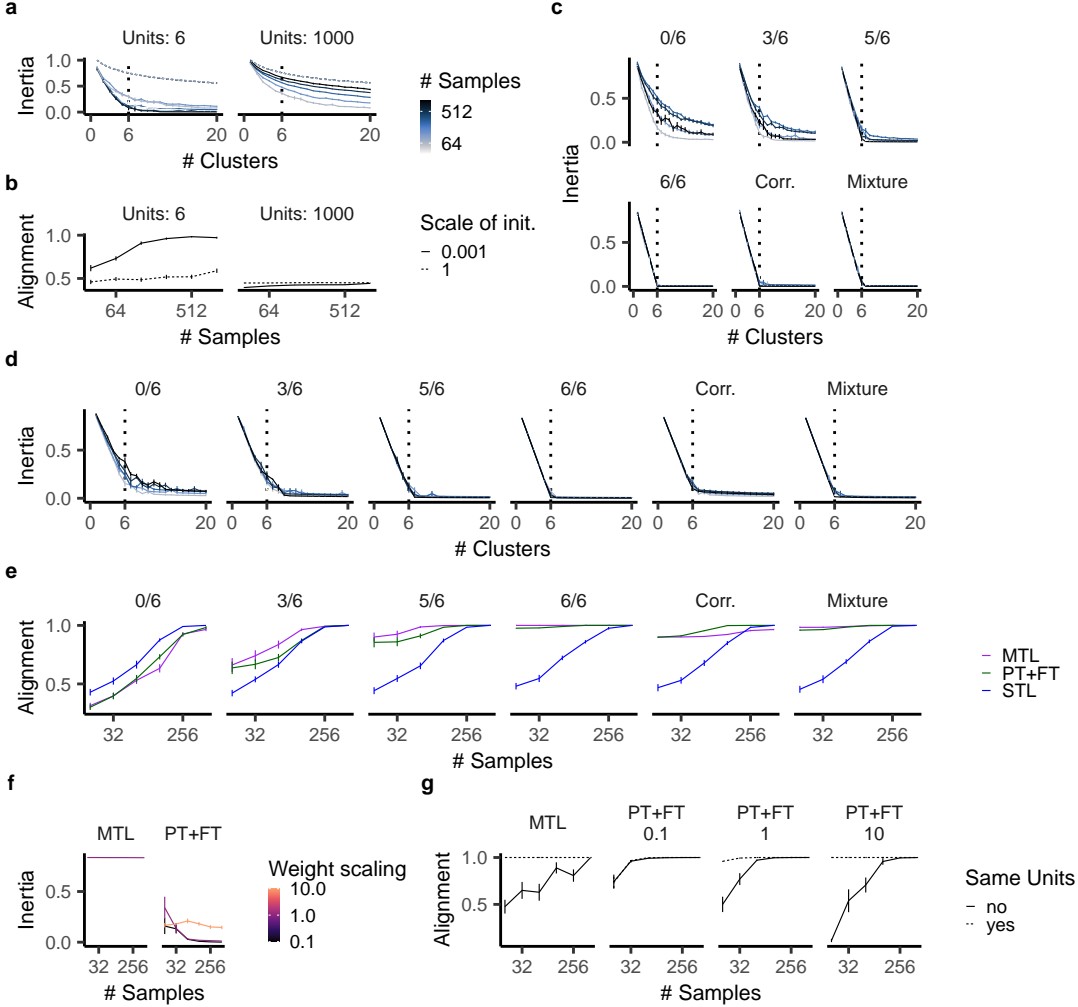

Figure 8: Analysis of effective sparsity of learned ReLU network solutions. **a** Inertia (k-means reconstruction error for clustering of hidden-unit input weights) as a function of the number of clusters used for $k$-means, for different numbers of main task samples and ground-truth teacher network units, in single-task learning. **b** Alignment score – average alignment (across teacher units) of the best-aligned student network cluster uncovered via k-means. **c**, Inertia for networks trained using PT+FT for the tasks of Fig. 2d,e and Fig. 4a. **d**, Same as panel $c$ but for networks trained with MTL. **e**, Alignment score for networks trained with MTL, PT+FT, and STL on the same tasks as in panels $c$ and $d$. **f** Inertia (using $k = 1$ clusters) for networks trained on an auxiliary task that relies on only one ground-truth feature, which is one of the six ground-truth features used in the auxiliary task (as in Fig. 3e,f), using MTL or PT+FT with various rescaling factors applied to the weights prior to finetuning. **g** Alignment score for the networks and task in panel $f$.

limited (Fig. 8e). This provides a mechanistic underpinning for the relationship between the inductive bias of PT+FT that we describe in the main text and its performance benefits.

### D.2.3 Nested feature selection regime allows network to prioritize a sparse subset of feature clusters learned during pretraining

In the main text we describe the "nested feature selection" regime, which occurs at intermediate values of the ratio between ground-truth main task feature coefficients and pretrained network weight scale. In this regime, networks can more efficiently learn main tasks that make use of a subset of the features used in the auxiliary task. Importantly, they still maintain a bias towards reusing features from the auxiliary task, as we show for diagonal linear networks and ReLU networks in Fig. 9. Here

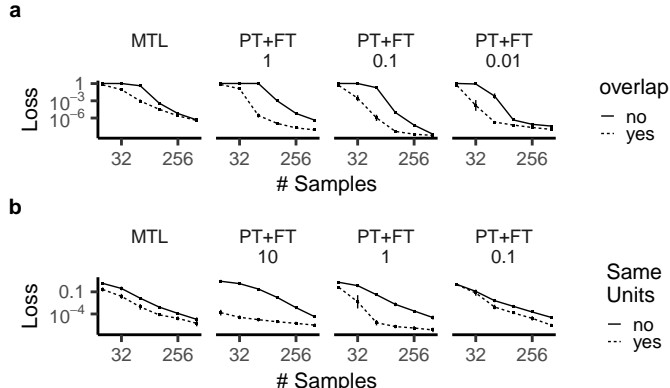

Figure 9: Comparison between tasks with sparse main task features that are either subsets of the auxiliary task features or new features. Networks are trained with MTL or with PT+FT, potentially with rescaling (as indicated by the number). **a**, Diagonal linear networks trained on five main task features. **b**, ReLU networks trained on a teacher network with one feature. We see that MTL (to some extent) and PT+FT can benefit from such an overlap, but for small rescaling values, this benefit becomes smaller.

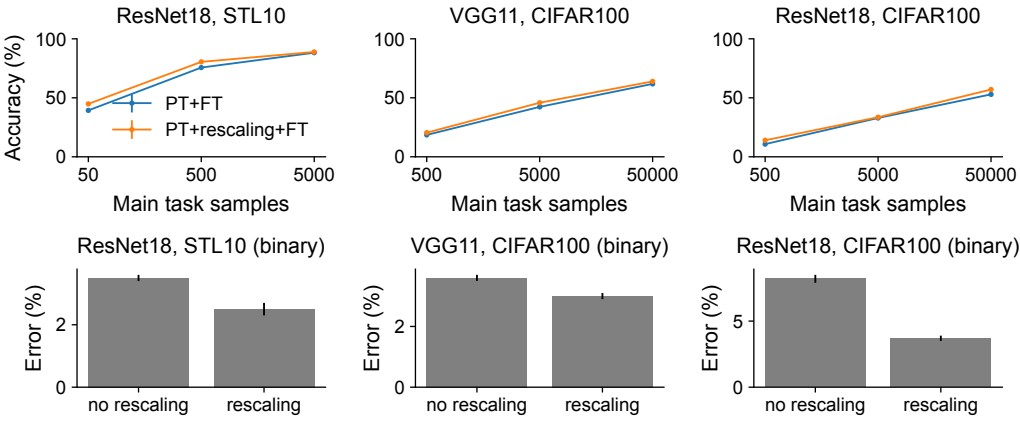

Figure 10: Results for finetuning deep convolutional networks trained on ImageNet, with/without weight rescaling (factor of 0.5) prior to finetuning.

we show that networks in this regime (obtained most clearly in the shallow ReLU network case when networks are rescaled by a value of 1.0 after pretraining) indeed learn very sparse (effectively 1-feature) solutions when the ground-truth main task consists of a single auxiliary task features (Fig. 8e, right). By contrast, networks with weights rescaled by a factor of 10.0 following pretraining exhibit no such nested sparsity bias (consistent with lazy-regime behavior). Similarly, multi-task networks cannot exhibit such a bias in their internal representation as they still need to maintain the features needed for the main task (Fig. 8e, left). Additionally, supporting the idea that the nested feature selection regime allows networks to benefit from feature reuse, we find that networks in this regime exhibit a higher alignment score with the ground-truth teacher network when the main task features are a subset of the auxiliary task features compared to when they are disjoint from the auxiliary task features (Fig. 8g). This alignment benefit is mostly lost when networks are rescaled by a factor of 0.1 following pretrainning (a signature of rich-regime-like behavior).

# E Further evaluations of the rescaling method for finetuning

To evaluate the robustness / general-purpose utility of our suggested approach of rescaling network weights following pretraining, we experimented with finetuning convolutional networks pretrained on ImageNet on downstream classification tasks: pretrained ResNet-18 finetuned on CIFAR100, pretrained VGG11 finetuned on CIFAR100, and pretrained ResNet-18 finetuned on STL-10. We experimented both with finetuning on the full multi-way classification task, and also on binary classification tasks obtained by subsampling pairs of classes from the main task dataset (which we found exposes performance differences more strongly). Due to computational constraints, we did not sweep over the choice of the rescaling factor, but simply used a factor of 0.5 in all cases. We find that rescaling improves finetuning performance, to varying degrees, in all of our experiments (Fig. 10).

# F Analysis of representations learned in the nested feature selection regime: bridging the gap from shallow to deep networks

We were interested in testing whether our theoretical understanding of shallow networks is truly responsible for the behavior of deeper networks (with more direct evidence than performance results / sample complexity). Specifically, we sought to understand whether the observed benefit of rescaling network weights following pretraining (Fig. 5b, Appendix E) relates to the nested feature selection regime we characterized in shallow networks. Doing so is challenging, as the space of "features" learnable by a deep network is difficult to characterize explicitly (making the feature clustering analysis employed in Appendix D inapplicable). To circumvent this issue, we propose a signature of nested feature selection that can be characterized without knowledge of the underlying feature space. Specifically, we propose to measure (1) the *dimensionality* of the network representation pre- and post-finetuning, and (2) the extent to which the representational structure post-finetuning is shared with / inherited from that of the network following pretraining prior to finetuning.

We employ the commonly used *participation ratio* [PR; 52] as a measure of dimensionality. For an $n \times p$ matrix $\mathbf{X}$ representing $n$ mean-centered samples of p-dimensional network responses, with a $p \times p$ covariance matrix $\mathbf{C}_X = \frac{1}{n}\mathbf{X}^\top\mathbf{X}$, the participation ratio is defined as

$$PR(X) = \frac{\left(\sum_{i=1}^p \lambda_i\right)^2}{\sum_{i=1}^p \lambda_i^2} = \frac{trace\left(\mathbf{C}_X\right)^2}{trace\left(\mathbf{C}_X^2\right)}$$
$$= \frac{trace\left(\mathbf{X}^\top\mathbf{X}\right)^2}{trace\left(\mathbf{X}^\top\mathbf{X}\mathbf{X}^\top\mathbf{X}\right)} \quad (36)$$

where $\lambda_i$ are the eigenvalues of the covariance matrix $\mathbf{C}_X$. The PR scales from 1 to $p$ and measures the extent to which the covariance structure of responses $\mathbf{X}$ is dominated by a few principal components or is spread across many. We argue that low-dimensional representations are a signature of networks that use a sparse set of features. We confirm that this is the case in our teacher-student setting: networks in the rich regime, which are biased towards sparse solutions, learn representations with lower PR than networks in the lazy regime, which are not biased toward sparse solutions (Fig. 11a).

Our measure of shared dimensionality between two representations is the *effective number of shared dimensions* (ENSD) introduced by Giaffar *et al.* [53]. The ENSD for an $n \times p$ matrix of responses $\mathbf{X}$ from one network and an $n \times p$ matrix of responses $\mathbf{Y}$ from another network, denoted $ENSD(X,Y)$, is given by

$$\frac{trace\left(\mathbf{Y}^\top\mathbf{X}\mathbf{X}^\top\mathbf{Y}\right) \cdot trace\left(\mathbf{X}^\top\mathbf{X}\right) \cdot trace\left(\mathbf{Y}^\top\mathbf{Y}\right)}{trace\left(\mathbf{X}^\top\mathbf{X}\mathbf{X}^\top\mathbf{X}\right) \cdot trace\left(\mathbf{Y}^\top\mathbf{Y}\mathbf{Y}^\top\mathbf{Y}\right)} \quad (37)$$

This measure is equal to the centered kernel alignment (CKA), a measure of similarity of two network representations [56], multiplied by the geometric mean of the participation ratios of the two representations. It measures an intuitive notion of "shared dimensions" — for example, if $\mathbf{X}$ consists of 10 uncorrelated units, if $\mathbf{Y}$ is taken from a subset of five of those units, the ENSD(X, Y) will be 5.

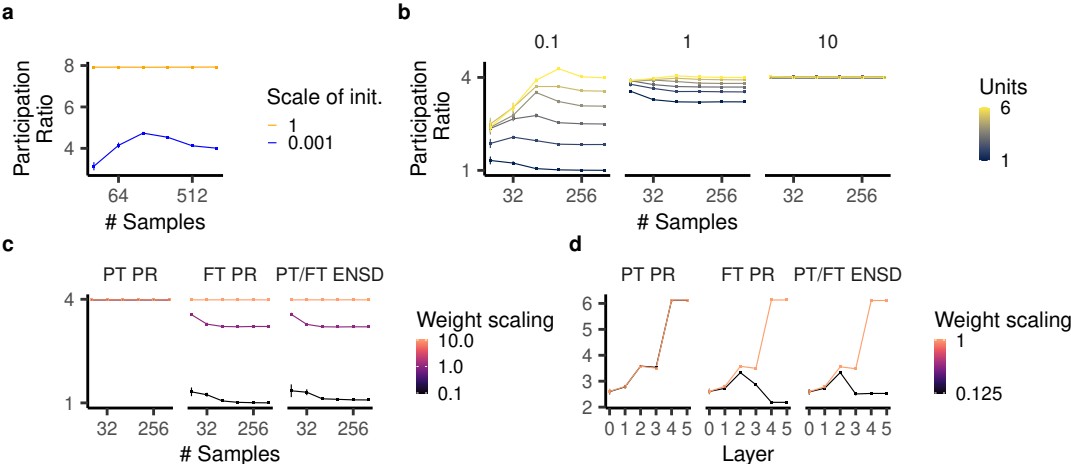

Figure 11: Dimensionality of the network representations before and after finetuning. **a**, Participation ratio of the ReLU networks' internal representation after training on a task with six teacher units. **b**, Participation ratio of the network representation after finetuning on the nested sparsity task with different weight rescalings. **c**, Participation ratio before (left panel) and after finetuning (middle panel) and the effective number of shared dimensions between the two representations (right panel). Small weight scaling decreases the participation ratio after training. **d**, The same quantities for ResNet18 before and after finetuning (see also Fig. 5c).

If $\mathbf{Y}$ is taken to be five uncorrelated units that are themselves uncorrelated with all those in $\mathbf{X}$, the ENSD(X, Y) will be zero.

Intuitively, the PR and ENSD of network representations pre- and post-finetuning capture the key phenomena of the nested feature selection regime. In a case in which the main task uses a subset of the features of the auxiliary task, if the network truly extracts this sparse subset of features, we expect the dimensionality of network after finetuning to be lower than after pretraining ($PR(\mathbf{X}_{FT}) < PR(\mathbf{X}_{PT})$), and for nearly all of the representational dimensions expressed by the network post-finetuning to be inherited from the network state after pretraining ($ENSD(\mathbf{X}_{PT}, \mathbf{X}_{FT}) \approx PR(\mathbf{X}_{FT})$). By contrast, networks not in the nested feature selection regime should exhibit an $\ell_2$-like rather than $\ell_1$-like bias with respect to features inherited from pretraining and thus not exhibit a substantial decrease in dimensionality during finetuning.

We show that this description holds in our nonlinear teacher-student experiments. Networks that we identified as being in the "nested feature selection" regime (weights rescaled by 1.0 following pretraining), and also networks in the rich regime, exhibit decreased PR following finetuning (Fig. 11b). By contrast, lazy networks (weights rescaled by 10.0 following pretraining) exhibit no dimensionality decrease during finetuning. Additionally (see Fig. 11c), the ENSD between pretrained (PT) and finetuned (FT) networks is almost identical to the dimensionality of the finetuned representation (PR FT).

Strikingly, we observe very similar behavior in our ResNet-18 model pretrained on 98 CIFAR-100 classes and finetuned on the 2 remaining classes (Fig. 11d), when we apply our method of rescaling weights post-finetuning. Analyzing the PR and ENSD of the outputs of different stages of the network following pretraining and following finetuning, we see that dimensionality decreases with finetuning, and ENSD between the pretrained and finetuned networks is very close to the PR of the finetuned network. Moreover, this phenomenology is only observed when we apply the weight rescaling method; finetuning the raw pretrained network yields no dimensionality decrease. These results suggest that the success of our rescaling method may indeed be attributable to pushing the network into the nested feature selection regime.

