# OpenReview forum: "Inductive biases of multi-task learning and finetuning: multiple regimes of feature reuse"
_NeurIPS.cc/2024/Conference — NeurIPS 2024 poster_

### Official Review · Reviewer_4YZg · 2024-07-07

**Soundness:** 3
**Presentation:** 1
**Contribution:** 2
**Rating:** 5
**Confidence:** 3

**Summary:**

This work investigates the implicit bias of both multi-task learning (MTL) and fine tuning pre-trained models (PT+FT) in diagonal linear networks and shallow ReLU neural networks. The contributions are as follows:
1. The authors prove that both MTL and PT+FT bias the networks to *reuse features*. This holds for both diagonal linear networks and shallow ReLU networks (Corollary 1, Corollary 2 and Proposition 3).
2. The authors prove that during finetuning networks interpolate between a "lazy" and "rich" regime. (Proposition 4).
3. For PT+FT it is observed empirically that the choosen features are a sparse subset of the features learned during pretraining.
4. The paper provides empirical evidence that for ReLU networks only PT+FT benefits from features that are correlated between the auxilary and primary task.
5. Finally, they present a practical technique to improve PT+FT by weight rescaling after pretraining.

**Strengths:**

I think the setting is interesting and very relevant to modern deep learning. This is also one of the few papers studying the inductive bias of multi-task networks. In particular:
- The paper offers a number of results with many experiments on toy models.
- The finding that MTL and PT+FT induces a regularizer that interpolates between $\ell_1$ and $\ell_2$ regularization is interesting and novel.
- The authors go beyond theory and show how their finding can be exploited in practical architectures through a simple weight rescaling technique.

**Weaknesses:**

- Corollary 1 and its extension for ReLU neural networks in Appendix A.1 is already well known in the literature [2] (see discuss after Remark 2) and [3] (Equation 4). These and the references therein should at least be mentioned.
- The works of [1] and [2] are both very related to the phenomenon uncovered here. In particular the notion of "neuron sharing" [2] seems to exactly correspond to the "feature reuse" phenomemon described in this work.
- It's not clear to me why the paper focuses on linear diagonal networks at all. These are not used in practice and are only studied for theoretical reasons. Since the theoretical contributions seem to be corollaries of previous results I think the paper would benefit from just focusing on shallow ReLU networks.
- The regularizer used during finetuning in Proposition 3 isn't well motivated. Minimizing the $\ell_2$ norm of the weight changes during fine tuning is not weight decay and seems a bit ad-hoc. It would be great to provide some discussion on why you choose to analyze this regularizer or whether this is done in practice.
- The notation is very loaded and makes the paper difficult to parse. For example $\vec{w}^{(1),aux}_{h}$.
- The figures are also very small and hard to read. It would be better to fill in all the whitespace between the different plots and include a legend for the different settings instead of a colorbar.


**Minor things**
- Lines 51 and 54 reuse the word "Finally".
- Line 82: Should the inputs to the model $x \in \mathbb{R}^{D}$?
- Line 113 paraeter --> parameter
- Line 135: Typo $\mathbb{R}^{d} \in \mathbb{R}$
- Line 251: yon --> on
- In Section 3 $D$ and $d$ are both used interchangeably for the input dimension of the networks.

[1] Collins, Liam, et al. "Provable multi-task representation learning by two-layer relu neural networks." arXiv preprint arXiv:2307.06887 (2023).

[2] Shenouda, Joseph, et al. "Variation Spaces for Multi-Output Neural Networks: Insights on Multi-Task Learning and Network Compression." arXiv preprint arXiv:2305.16534 (2023).

[3] Yang, Liu, et al. "A better way to decay: Proximal gradient training algorithms for neural nets." OPT 2022: Optimization for Machine Learning (NeurIPS 2022 Workshop). 2022.

**Questions:**

- Where is the proof for Corollary 2?
- When finetuning are the weights of both layers trained or only the second layer weights?
- Where are the results from the experiments discussed in Section 3.3?
- In Fig 2a, what do the labels on the colorbars mean "# Active dims" and "Units"? Is this the number of features in the teacher model?
- Line 176 what do you mean by "overlapping dimensions"? You mean overlapping features?
- On line 178 what is LP in PT+FT(LP)?

**Limitations:**

Limitations have been mentioned in the weaknesses section.

---

> ### Author Rebuttal · Authors · 2024-08-06
>
> Thank you very much for your helpful review. Below we respond to your comments and questions. (Due to the character limit on the rebuttal, we focus here on the most important responses and have relegated additional responses to a comment.)
>
> First, in response to the reviewer’s summary of the paper, we wanted to clarify a few of our key contributions. If you think we could highlight these contributions more clearly in the revised version of the paper, we would appreciate any suggestions.
>
> > The authors prove that during finetuning networks interpolate between a "lazy" and "rich" regime. (Proposition 4).
>
> Specifically, we show that this interpolation can be described in terms of a conserved quantity expressing a tradeoff between initialization dependence and sparsity, and that a network’s position on this tradeoff is tied to the scale of the weights following pretraining.
>
> > For PT+FT it is observed empirically that the choosen features are a sparse subset of the features learned during pretraining.
>
> The insight above allowed us to predict the existence of a “nested feature selection” regime, in which finetuning extracts a sparse subset of the features learned during pretraining (due to intermediate levels of both sparsity and initialization dependence). This is a qualitatively different regime from the lazy regime (which is initialization-dependent, but not sparse) and rich regime (which is sparse, but not initialization-dependent) that were known previously. Networks do not always exhibit this behavior, but do in some cases, and can be induced to exhibit it via the weight rescaling technique we introduce.
>
> **Responses to comments**
>
> > - Corollary 1 and its extension for ReLU neural networks in Appendix A.1 is already well known in the literature [2] (see discuss after Remark 2) and [3] (Equation 4). These and the references therein should at least be mentioned.
>
> > - The works of [1] and [2] are both very related to the phenomenon uncovered here. In particular the notion of "neuron sharing" [2] seems to exactly correspond to the "feature reuse" phenomemon described in this work.
>
> Thank you for pointing us to these highly relevant papers; they were interesting to read and indeed highly relevant. We’ll make sure to cite these papers when describing the MTL penalty and when describing the feature reuse bias (in particular, Theorem 9 in [2]). We will also discuss this line of work in the Related Work section. We note that in our original draft we did not claim that Corollary 1 was an original result (we cited reference [16], Dai et al.) but we appreciate the relevant references for the ReLU network case and agree they are important to cite.
>
> > It's not clear to me why the paper focuses on linear diagonal networks at all. These are not used in practice and are only studied for theoretical reasons. Since the theoretical contributions seem to be corollaries of previous results I think the paper would benefit from just focusing on shallow ReLU networks.
>
> We see our results on diagonal linear networks as complementary to our results on ReLU networks, as we are able to prove our theoretical claims with fewer assumptions in the diagonal linear case.  In particular, the diagonal linear setting allows us to analytically derive the effects of implicit regularization due to gradient descent for PT+FT (in the ReLU case we had to assume an explicit regularizer, due to the difficulty of analyzing gradient descent dynamics in nonlinear networks). This helps clarify the mechanism underlying our key theoretical results -- the nested feature-selection regime and our “conservation law” -- and motivates our subsequent ReLU network analysis.
>
> > The regularizer used during finetuning in Proposition 3 isn't well motivated. (...) It would be great to provide some discussion on why you choose to analyze this regularizer or whether this is done in practice.
>
> We agree that this deserves discussion separately from the explicit regularization considered in the multi-task learning setup. We chose to consider this regularization penalty for two reasons. First, infinitesimal explicit regularization from initialization is equivalent to the implicit regularization induced by gradient descent in the case of shallow linear models [4 below]. While this is not true more generally, it is a useful heuristic to motivate theoretical analysis (which of course must then be checked against experiments). Assuming the explicit regularization heuristic allowed us to derive a number of nontrivial predictions that we would not otherwise have come up with, and which we confirmed hold true in empirical simulations of unregularized gradient descent.  We also note that this explicit regularization penalty is sometimes studied in the context of continual learning [e.g. 5,6 below].  We will make sure to clarify the motivation for basing our ReLU theory on this regularization penalty in the revised manuscript.
>
> > Where is the proof for Corollary 2?
>
> This was an oversight on our part. We will add an explanation to the appendix detailing how this follows from Azulay et al. Specifically, we derive this results in two steps: first, we note that if after pretraining the network has the effective linear predictor $\beta^{aux}$, the first hidden layer has the weights $\sqrt{\beta^{aux}}$, where the square root is applied element-wise. Having set the readout initialization to $\gamma$, we then apply Theorem 4.1 in Azulay et al.
>
> > Where are the results from the experiments discussed in Section 3.3?
>
> Section 3.3 describes the setup for all the teacher-student experiments presented in section 4. We will clarify this in the revised manuscript.
>
> 4. Gunasekar et al. "Characterizing implicit bias in terms of optimization geometry." ICML (2018).
> 5. Lubana et al. "How do quadratic regularizers prevent catastrophic forgetting: The role of interpolation." CoLLAs (2022).
> 6. Evron et al. "Continual learning in linear classification on separable data." ICML (2023).

---

> ### Author Response · Authors · 2024-08-07
> **Additional responses**
>
> Below are additional responses to your questions and suggestions that we did not have sufficient space for in the main rebuttal.
>
> > - The notation is very loaded and makes the paper difficult to parse. For example $w^{(1),aux}_h$.
> > - The figures are also very small and hard to read. It would be better to fill in all the whitespace between the different plots and include a legend for the different settings instead of a colorbar
>
> Thank you for highlighting this. We’ll change the notation to denote hidden weights (i.e. $w^{(1)}$) by $w$ and readout weights (i.e. $w^{(2)}$) by $v$, in addition to a few other changes that aim to clarify notation a bit. We also appreciate the suggestion on the figures — we’ll make them bigger and agree that replacing some of the color bars by legends would be helpful.
>
> > When finetuning are the weights of both layers trained or only the second layer weights?
> > On line 178 what is LP in PT+FT(LP)?
>
> Finetuning means that both layers are trained whereas LP stands for linear probing, meaning that only the readout weights are trained. Importantly, this means that finetuning is able to perform feature learning whereas linear probing cannot. We will add a sentence clarifying this to the manuscript.
>
> > In Fig 2a, what do the labels on the colorbars mean "# Active dims" and "Units"? Is this the number of features in the teacher model?
>
> Yes, exactly. Would “# Non-zero dims.” and “# Units” be clearer?
>
> > Line 176 what do you mean by "overlapping dimensions"? You mean overlapping features?
>
> Yes. We will change this to “overlapping non-zero dimensions (i.e. features)” (as we want to make clear that in this case, the different features are simply different dimensions of the input).
>
> Again, thank you very much for your review!

---

> ### Comment · Reviewer_4YZg · 2024-08-11
> **Thank you for the response**
>
> I thank the authors for their rebuttal which have clarified many of my questions and addressed the concerns raised. While I still have some reservations about the presentation, I think the results are interesting and I trust the authors will make the necessary changes in the camera-ready, therefore I will raise my score to 5 (Borderline Accept).
>
> Aside: I still think that while the results on diagonal linear networks are neat and clean they don't really seem necessary and they distract from some of the more interesting points in the paper. I would really suggest moving all of the diagonal linear net stuff like Corollary 1/2 and diagonal linear experiments to a section in the Appendix. I would use the extra space to make these figures bigger, provide more justification on some of the assumptions being used in Prop. 3 and move some of the experiments from the Appendix to the main body.
>
> **Typos**
> - Line 87 and 88: What is $\ell_1$ and $\ell_2$ norm of $f$ are these typos? It seems like you're taking an $\ell_2$ norm of a function which doesn't really make sense...
> - Line 85 has a switch in the citation style.

---

> ### Comment · Reviewer_969G · 2024-08-11
> **A Small Contrasting Opinion Re Linear Diagonal Networks**
>
> I want to note that I differ slightly from Reviewer 4YZg on the recommendation about moving linear diagonal network results to Appendix -- I think as the authors say, including the results gives an insight starting from strong theoretical backing with limited assumptions that they then show in increasingly realistic settings, albeit with less formalism. As a theorist, I value the range of models in which this insight is presented; it not only substantiates the authors' results from multiple epistemic perspectives but also is encouraging to theorists to see that the simple models we study corroborate insights derived via different tools (importantly, in different settings than in which the model was initially developed)! Including these results in the main body only strengthens the narrative, in my opinion.
>
> I could imagine that for an empiricist reading this paper, the results in the extremely stylized model may not not provide as much value. I encourage the authors to think about their target audience(s) in making the decision of whether or not to move the results to the appendix.

---

> > ### Comment · Reviewer_FiYt · 2024-08-12
> > **Seconding the opinion to keep diagonal network in main paper not Appendix**
> >
> > I will second the opinion of Reviewer 969G here. My initial impression was the same as Reviewer 4YZg on the use of linear diagonal networks as the initial proof of concept. My main issue is that these are very contrived networks with specific properties that are not used at all in contemporary deep learning. However, the more I read about them the more I saw that they still act as a stepping stone in theoretical studies and are an excellent starting point for explainable deep learning re: feature reuse. After reading the entire paper several times I think the LDN section has a place in the main paper not appendix.

---

> ### Author Response · Authors · 2024-08-13
>
> We are glad that our rebuttal addressed the reviewer's concerns and thank them for increasing their score.
>
> We also appreciate their note on the diagonal linear network section (suggesting that it should be moved to the appendix), as well as the input on this question by reviewers 969G and FiYT (who suggest it is a useful part of the main article, for the reason that we provide in the rebuttal). We really appreciate everyone's input on this. We have decided to keep the diagonal linear network section in the main text. However, we will clarify in more detail the role of these networks in connecting the theory in our paper to existing lines of work in deep learning theory. Note that in response to a suggestion by reviewer Ry6c, we will move our definitions of the diagonal linear and ReLU networks to a dedicated section ("Theoretical setup") before describing our theoretical results. We will add to this section a detailed explanation of the role of diagonal linear networks in our paper.
>
> We believe that this will make our paper more broadly accessible: for readers who are interested in these networks as a theoretical stepping stone, we present diagonal linear networks in an integrated manner with our other findings; for empirically minded readers, we clarify early on why we focus on these networks and make it as easy as possible for them to focus their attention on ReLU networks and large-scale neural networks (if they wish to do so). We will also make sure to provide more justification on the assumptions used in Prop. 3 and make our figures bigger (using the extra page allotted to the camera-ready version).
>
> We want to thank all the reviewers for providing their input on this question, as this was a very helpful discussion.
>
> **Response to typos**
>
> > Line 87 and 88: What is $\ell_1$ and $\ell_2$ norm of $f$ are these typos? It seems like you're taking an $\ell_2$ norm of a function which doesn't really make sense...
>
> Thank you for pointing this out. Our intention in these lines was to define the $\ell_1$/$\ell_2$-norm of these functions in terms of the $\ell_1$/$\ell_2$-norm of the linear coefficients (as we consider linear functions). However, we agree that this is confusing and will replace $\|f\|_{\ell_1}$/$\|f\|_{ell_2}$ by $\|\beta\|_1$/$\|\beta\|_2$.
>
> > Line 85 has a switch in the citation style.
>
> Thanks for pointing this out.
>
> Once again, thank you very much for your response as well as your original review.

---

### Official Review · Reviewer_FiYt · 2024-07-08

**Soundness:** 3
**Presentation:** 3
**Contribution:** 4
**Rating:** 7
**Confidence:** 3

**Summary:**

Abstract

The goals of the paper are very clear from the abstract, as are the results.

Introduction

Lines 23-42: The authors do a great job of summarizing the applications of MTL (using it here loosely to capture MTL and PF+FT) while recognizing that very little work has been done to explore exactly why it works as well as it does in data-limited settings. We have strong intuitions about its regularization effects with minimal work backing our intuitions

Related Work
Lines 60-68: Noting the extent of prior work analyzing regularization effects in single-task modeling is hopeful in showing the impact of the author’s work in adding to similar work for MTL.

Lines 85-89: I would do a better job of defining “large initialization” vs “small initialization” for the reader without requiring a reference check by the reader. One could assume that large = overparameterized and small = underparameterized versus large vs small magnitudes for the same number of parameters at initialization.

Implicit and explicit regularization penalties for MTL and PT+FT

Lines 94-102: I think most readers would take the authors at their heuristic given what they note in Lines 101-102, that explicit weight decay use in practice yields a permissible heuristic. However, the references, especially to [12] enhance their argument.

Lines 103-111: The notation in Corollary 1 is a bit confusing. I am not sure why beta vector aux has the aux not in subscript when defining vector beta 2. I also do not know why we introduce vector beta 2 other than just showing that we are indexing the outputs since we continue to stick with vector beta aux. I see that in 3 we now subscript the dimension but I think the notation can be more clean.

Line 112: Typo: “A ReLU networks”

Lines 112-115: I have not seen a ReLU network defined this way and from the authors’ text the notation is again confusing. The authors state they are defining a ReLU network with O outputs but (4) that follows defines only a single final output from the network. What follows then makes us think that the “outputs” are the units in a single hidden layer? The notation is unnecessarily confusing for a basic ReLU network and something more conventional would help especially in the context of MTL. Also, the L1,2 norm is referenced several times already but not defined. There is another typo on line 113 “paraeter”.

Lines 117-124: Why are the DLN network weights initialized with a constant magnitude?

Lines 128-129: We are not told how we get to (6) and not appendix section is referenced

Line 131: Typo

Line 134: Herein we find more confusion because now (4) is referenced as an equation for a ReLU network with a single output, whereas this was not how it was defined in the preceding text before Equation (4). Also, weights within the set of R d within the set of R?

Line 135: Is vector gamma simply a uniform vector (constant magnitude) of the same scalar similar to how the DLN was defined or is the vector gamma different values at re-initialization (not uniform)?

Lines 137-139: It is not until Line 139 that theta and m are well defined despite numerous equations/references to them earlier in the manuscript.

Figure 2: I would define STL as single task learning for the less familiar reader

Lines 144-151: The experiment is explained well and easy to follow.

Lines 152-155: More explanation for the ReLU would help. By “sparse” number of (hidden?) units we mean a low dimensional hidden layer?

Lines 158-165: Everything is clear save how we arrived at c. Appendix reference?

Lines 158-182: I like the elegance of the intuition to simplify the experiments but just having no feature overlap in aux and main tasks in small pretrained feature case and full feature overlap in aux and main in large pretrained feature case.

Lines 183-194: Again, elegance in defining the feature sharing to explicitly study what was of interest (simultaneous sparsity and feature sharing).

Lines 206-215: The definitions are clear and the intermediate regime is set up well by the authors.

Lines 223-250: Again, the elegance of the experiment set-ups to reinforce the nested feature selection regime.

Lines 256-257: I would like to hear more about this from the authors in the Conclusion. It seems like the authors are saying that there seems to be some “critical” (used loosely here) rescaling magnitude between aux aux and main tasks in MTL over which the network is unable to enter the nested feature selection regime. I would have liked to see the authors stress this to see what they found re: network tolerance to different rescaling magnitudes. Hopefully this can be a future work.

Lines 262-264: I am not sure this is the case. MTL can certainly learn correlated features in shallow layers where one task’s use of the common features are simply rescaled later in the network relative to the features learned for an auxiliary task in isolation. Unless the authors are speaking more strictly of actual soft parameter sharing here, in which case, yes, separate parameters are being learned but kept similar. I think this is confirmed in the low sample MTL results (where we know MTL works best when it can). It does not work as well as PT+FT but still beats STL. It is a marginal beat.

Lines 312-315: Very interesting finding on a large dataset and more complex network.

Summary: The authors explore the inductive biases behind the success of MTL and PT+FT, a very under-studies phenomenon. Many interesting discoveries were made, most interestingly in their newly names "nested feature selection regime". This is important and timely work. I would like some of the notation cleaned-up for better reach to a wide audience.

**Strengths:**

The theoretical justifications for the set-up of the experiments are excellent. The experiments that follow are also well-designed to allow the authors to probe their implicit regularization penalties. The authors kept probing until arriving on their nested feature regime. I liked the trajectory from theory to synthetic data/systems experiments to findings to larger/classic datasets and networks.

**Weaknesses:**

I think some of the notation needs to be cleaned-up to have this important work reach as broad an audience as possible. Much is left to the appendix and when it is the references are usually there, but I would make sure they are always there for the reader to reference from the main body of the paper.

**Questions:**

Why are the DLN network weights initialized with a constant magnitude?

The authors mention how entrance into the nested feature regime is sensitive to the correlation between aux and main task features. Some more thoughts in the Conclusions about further theoretical thresholding of this or future experiments would be of interest to me as the authors seemed to have good theoretical foundation for all other experiments in the paper.

Beyond me being pedantic, is this entire paper working on multi-output learning not multi-task learning? Traditionally when we talk about MTL we have two tasks with a different loss function, different goals, different processes, different dimensionality of input features (image classification as task 1 and image object segmentation as task 2). From what I can tell, the auxiliary and main tasks in the synthetic experiments are different instantiations of the same process, with carefully crafted differences in features/coefficients/etc in order to generate different outputs, then described as different tasks. I would describe this as multi-output learning. All of the findings still hold of course, but I would make the distinction more clear.

---

> ### Author Rebuttal · Authors · 2024-08-06
>
> Thank you very much for your positive feedback and your helpful comments. We will make sure to make the notation more accessible and really appreciate your detailed notes on this. Below we respond to the questions and comments you raised in your review. (Due to the character limit, we focus on the most important responses in the rebuttal and have relegated additional responses to a comment.)
>
> > Why are the DLN network weights initialized with a constant magnitude?
>
> We wanted to remove the random variation induced by having randomly sampled readout weights, and  to be consistent with Woodworth et al. who also considered constant-magnitude initialization. However, we would expect qualitatively similar results for randomly sampled weights from a Gaussian. Unlike for ReLU networks, random weights are not really necessary for symmetry breaking in diagonal linear networks, as each unit already receives input from a different dimension.
>
> > The authors mention how entrance into the nested feature regime is sensitive to the correlation between aux and main task features. Some more thoughts in the Conclusions about further theoretical thresholding of this or future experiments would be of interest to me as the authors seemed to have good theoretical foundation for all other experiments in the paper.
>
> We agree that the task dependence of where the nested feature selection regime falls is an interesting phenomenon.  As the reviewer points out, we show a theoretical basis for such sensitivity in Fig. 3d.  However, some of our empirical results are not completely described by the theory – empirically, even when auxiliary and main task features are identical, ReLU networks exhibit behavior that our theory would expect to arise in the case where they are highly correlated but not identical. We conjecture that this discrepancy arises due to noise in stochastic gradient descent dynamics (i.e. the inferred features on the two tasks may be highly correlated but not identical).  Furthermore, we observe differences between ResNets and Vision Transformers in where the nested feature selection regime occurs.  Our shallow network theory is not able to speak to these architectural differences. We think that clarifying this picture is an important direction for future work, and will add a sentence on this to the conclusion.
>
> > Beyond me being pedantic, is this entire paper working on multi-output learning not multi-task learning? (...) From what I can tell, the auxiliary and main tasks in the synthetic experiments are different instantiations of the same process, with carefully crafted differences in features/coefficients/etc in order to generate different outputs, then described as different tasks. I would describe this as multi-output learning. All of the findings still hold of course, but I would make the distinction more clear.
>
> We agree that in practice multi-task learning is much more versatile than the setup we consider here. We opted for such similar main and auxiliary tasks to remove additional factors of variation and to keep the teacher-student setup parsimonious, but will update the manuscript to acknowledge the much broader range of multi-task training used in practice. We expect that our findings are relevant to other kinds of multi-task learning, but have not yet established this, and think it is an important direction for future work.
>
> > Lines 256-257: I would like to hear more about this from the authors in the Conclusion. It seems like the authors are saying that there seems to be some “critical” (used loosely here) rescaling magnitude between aux aux and main tasks in MTL over which the network is unable to enter the nested feature selection regime. I would have liked to see the authors stress this to see what they found re: network tolerance to different rescaling magnitudes. Hopefully this can be a future work.
>
> We agree that this is a really fascinating phenomenon. So far, we have found a (not fully rigorous) theoretical basis for differences in the rescaling magnitude required to enter the nested feature selection regime between diagonal linear networks and ReLU networks (see lines 245-250 for our explanation). We have also empirically found that rescaling by a lower value is required for ResNets to enter the nested feature selection regime, whereas this is not required for Vision Transformers (and indeed such rescaling appears to cause them to leave this regime). We don’t yet understand the reason for this difference, and we agree that a more comprehensive investigation of this phenomenon (across networks and tasks) would be very interesting future work. We will add a sentence on this topic to the conclusion.
>
> > Lines 262-264: I am not sure this is the case. MTL can certainly learn correlated features in shallow layers where one task’s use of the common features are simply rescaled later in the network relative to the features learned for an auxiliary task in isolation. Unless the authors are speaking more strictly of actual soft parameter sharing here, in which case, yes, separate parameters are being learned but kept similar. I think this is confirmed in the low sample MTL results (where we know MTL works best when it can). It does not work as well as PT+FT but still beats STL. It is a marginal beat.
>
> To clarify, while MTL can certainly learn correlated features using different hidden units, it does not benefit from such correlated units in terms of its overall norm (at least in a neural network with one hidden layer). Note that in the low-sample regime, MTL simply uses the same unit for both tasks, which is not entirely accurate but works reasonably well (see Fig. 8e, which shows that the student units have a correlation of 0.9 with the main task teacher, suggesting that they are identical with the units the student uses for the auxiliary task). We'll change this sentence to make that point clearer.

---

> > ### Comment · Reviewer_FiYt · 2024-08-08
> > **First Response**
> >
> > I appreciate the authors' follow-up responses. Much of my questions were asking for clarity on theoretical implications of the authors' work. Although it is not all addressed in the work here, the authors have many ideas that they are adding into the Discussion and Conclusions sections to drive further work by them and others.
> > I think my point re: " I am not sure this is the case. MTL can certainly learn correlated features in shallow layers where one task’s use of the common features are simply rescaled later in the network relative to the features learned for an auxiliary task in isolation" came from the authors and myself referencing two different things. I was certainly referring to a NN with more than one layer whereas the authors were making their statement in the context of a NN with a single hidden layer. I do not think we have any disagreement in the more restricted case being described by the authors.

---

> > > ### Author Response · Authors · 2024-08-13
> > >
> > > Thank you for your response, and thank you again for your helpful review. We agree that MTL could encode correlated features in a network with more than one hidden layer, which could be an interesting direction for future work.

---

> ### Author Response · Authors · 2024-08-07
> **Additional responses**
>
> Below, we are providing some additional responses that we did not have space for in our rebuttal:
>
> > Lines 85-89: I would do a better job of defining “large initialization” vs “small initialization” for the reader without requiring a reference check by the reader.
>
> Thank you for pointing this out! We will change "large/small initialization" to "large/small initial weight magnitude".
>
> > Lines 103-111: The notation in Corollary 1 is a bit confusing. I am not sure why beta vector aux has the aux not in subscript when defining vector beta 2. I also do not know why we introduce vector beta 2 other than just showing that we are indexing the outputs since we continue to stick with vector beta aux. I see that in 3 we now subscript the dimension but I think the notation can be more clean.
>
> We agree that the notation is overly complicated and will clean this up. Specifically, we will now introduce the two output dimensions by the superscripts “aux” and “main” from the beginning.
>
> > Lines 112-115: I have not seen a ReLU network defined this way and from the authors’ text the notation is again confusing. The authors state they are defining a ReLU network with O outputs but (4) that follows defines only a single final output from the network. What follows then makes us think that the “outputs” are the units in a single hidden layer? The notation is unnecessarily confusing for a basic ReLU network and something more conventional would help especially in the context of MTL. Also, the L1,2 norm is referenced several times already but not defined. There is another typo on line 113 “paraeter”.
>
> > Lines 137-139: It is not until Line 139 that theta and m are well defined despite numerous equations/references to them earlier in the manuscript.
>
> Thank you for highlighting this! We’ll make the notation here more conventional and will add a separate remark early on explaining the alternative parameterization by the magnitude $m$ and unit direction $\theta$ (as this parameterization is important for characterizing the penalties we derive). We’ll also make sure to properly define the L1,2 norm.
>
> > Lines 128-129: We are not told how we get to (6) and not appendix section is referenced
>
> This was an oversight on our part. We will add an explanation to the appendix detailing how this follows from Azulay et al. Specifically, we derive this results in two steps: first, we note that if after pretraining the network has the effective linear predictor $\beta^{aux}$, the first hidden layer has the weights $\sqrt{\beta^{aux}}$, where the square root is applied element-wise. Having set the readout initialization to $\gamma$, we then apply Theorem 4.1 in Azulay et al. Note that we use a slightly different functional form of $q$ to enhance readability.
>
> > Line 134: Herein we find more confusion because now (4) is referenced as an equation for a ReLU network with a single output, whereas this was not how it was defined in the preceding text before Equation (4). Also, weights within the set of R d within the set of R?
>
> To clarify, the single-output ReLU network is obtained by setting $O=1$ in Eq. 4. However, as noted above, we will change our explanation of this setup to make it clearer. The "$\in\mathbb{R}$" is a typo; thank you for pointing it out.
>
> > Line 135: Is vector gamma simply a uniform vector (constant magnitude) of the same scalar similar to how the DLN was defined or is the vector gamma different values at re-initialization (not uniform)?
>
> Our theorem applies to arbitrary vectors $\gamma$. In practice we use a randomly sampled readout with a variance of $10^{-3}\sqrt{2/H}$. We will add a sentence clarifying this point; thank you for pointing this out.
>
> > Figure 2: I would define STL as single task learning for the less familiar reader
>
> We agree.
>
> > Lines 152-155: More explanation for the ReLU would help. By “sparse” number of (hidden?) units we mean a low dimensional hidden layer?
>
> Yes, i.e. a low number of hidden units. We will clarify this --- thank you for pointing this out!
>
> > Lines 158-165: Everything is clear save how we arrived at c. Appendix reference?
>
> We should have clarified that this is a direct application of an analysis in Woodworth et al., and will add such a clarification.
>
> > Lines 158-182: I like the elegance of the intuition to simplify the experiments but just having no feature overlap in aux and main tasks in small pretrained feature case and full feature overlap in aux and main in large pretrained feature case.
>
> > Lines 183-194: Again, elegance in defining the feature sharing to explicitly study what was of interest (simultaneous sparsity and feature sharing).
>
> > Lines 223-250: Again, the elegance of the experiment set-ups to reinforce the nested feature selection regime.
>
> Thank you very much for these (and the other) positive comments! It’s useful for us to get this kind of feedback, as it helps us know when our explanations/experiments are clear.
>
> Again, thank you very much for your helpful review!

---

### Official Review · Reviewer_Ry6c · 2024-07-12

**Soundness:** 3
**Presentation:** 2
**Contribution:** 3
**Rating:** 5
**Confidence:** 2

**Summary:**

In this study, the authors explore the inductive biases associated with multi-task learning (MTL) and the sequential process of pretraining followed by finetuning (PT+FT) in neural networks. Specifically, they analyze the implicit regularization effects in diagonal linear networks and single-hidden-layer ReLU networks under these training paradigms. The findings reveal that both MTL and PT+FT promote feature reuse across tasks and favor sparsity in the feature set, establishing a conservation law that illustrates a tradeoff between these biases. Additionally, a unique "nested feature selection" pattern in PT+FT is identified, where a sparse subset of features learned during pretraining is selectively refined. This behavior contrasts with the broader feature learning strategies seen in MTL. Empirical validation using teacher-student models and experiments on deep networks trained for image classification confirms the theoretical insights. The authors also propose a practical enhancement involving weight rescaling post-pretraining, which their results suggest can optimize finetuning by encouraging the network to engage more effectively in the nested feature selection regime.

**Strengths:**

1. The article provides an in-depth characterization of the inductive biases associated with two common training strategies—Multi-Task Learning (MTL) and Pretraining followed by Fine-Tuning (PT+FT)—in diagonal linear and ReLU networks. This detailed analysis is crucial for understanding the impacts of different training strategies.

2. By pushing networks into the nested feature-selection regime, the article proposes simple techniques to improve PT+FT performance, which have shown promising empirical results, adding practical application value.

**Weaknesses:**

1. Although the findings are promising, the article notes that more work is needed to test these phenomena in more complex tasks and larger models. This suggests that the research's applicability and universality might be limited.

2. The article outlines promising avenues for extending the theoretical work, such as connecting derived penalties for ReLU networks to the dynamics of gradient descent, and extending the theory to the case of cross-entropy loss. This implies that the current theoretical foundation still requires further development.

3. The expression of the article needs further improvement. It is suggested that the author should introduce the theoretical settings of analyzing MTL and PT+FT in separate sections in detail, rather than the current form scattered across various sections.

**Questions:**

See in Weaknesses.

---

> ### Author Rebuttal · Authors · 2024-08-06
>
> Thank you very much for your helpful review. Below we respond in detail to your comments.
>
> > Although the findings are promising, the article notes that more work is needed to test these phenomena in more complex tasks and larger models. This suggests that the research's applicability and universality might be limited.
>
>
> We are indeed quite excited about investigating these phenomena in such more complex settings — we expect that such an investigation would reveal an even more nuanced picture in terms of when and how an inductive bias towards nested feature sparsity, correlated features, and shared features aids generalization. We believe that such an investigation is best conducted in a new manuscript, whereas the main focus of this paper was to carefully derive and explain how these inductive biases arise in simpler networks. We’d also like to emphasize that we’re already providing an investigation of these empirical phenomena that considers several deep neural network architectures (ResNet, VGG, Vision Transformers) and standard benchmark datasets (Imagenet and CIFAR-10). As such, we believe that we are already providing useful evidence for the applicability of our insights, but we acknowledge that there is much more work to be done on this in future papers.
>
> > The article outlines promising avenues for extending the theoretical work, such as connecting derived penalties for ReLU networks to the dynamics of gradient descent, and extending the theory to the case of cross-entropy loss. This implies that the current theoretical foundation still requires further development.
>
> We agree that these are exciting avenues. Understanding how the penalties arising from explicit regularization differ from and connect to the penalties arising from the implicit regularization of gradient descent is technically challenging but an important direction. (We note that characterizing the impact of explicit regularization is also useful and important in its own right, and is a common strategy for understanding the inductive biases of networks, see e.g. Savarese et al. (2019); Evron et al. (2022); see also l.97-102 in our manuscript.) Similarly, we would be interested in extending our theory to the cross-entropy loss (as well as other losses) and expect the same principles of feature sparsity, sharing, and correlations to be important in that setting as well. Doing so would require introducing new teacher-student setups and introduce some additional practical considerations (e.g. weights of cross-entropy networks grow without bound rather than converge) which we believe are best left for another paper. We felt it was better to focus this paper on analyzing the regression setting in depth, to provide a solid foundation for future extensions to our work.
>
> > The expression of the article needs further improvement. It is suggested that the author should introduce the theoretical settings of analyzing MTL and PT+FT in separate sections in detail, rather than the current form scattered across various sections.
>
> Thank you for this suggestion. We agree that describing the theoretical setup in a separate section could be a helpful change. As a result, we will make the following change to the camera-ready version if accepted: we will add a new section 3.1 (“Theoretical setup”), which defines diagonal linear networks and ReLU networks and defines multi-task learning and pretraining+finetuning. We will then introduce our theoretical results in dedicated sections.  Currently, our sections 3.1 and 3.2 both introduce the network setups and describe theoretical results – these changes will have the effect of refactoring these sections to separate our description of the setup from our theoretical results. Could you clarify if this is the kind of change you had in mind? We are open to other ways of organizing the content.
>
> **References**
>
> Savarese, Pedro, et al. "How do infinite width bounded norm networks look in function space?." Conference on Learning Theory. PMLR, 2019.
>
> Evron, Itay, et al. "How catastrophic can catastrophic forgetting be in linear regression?." Conference on Learning Theory. PMLR, 2022.

---

### Official Review · Reviewer_969G · 2024-07-13

**Soundness:** 4
**Presentation:** 3
**Contribution:** 4
**Rating:** 8
**Confidence:** 3

**Summary:**

This paper studies the implicit bias of gradient descent on the linear diagonal model and two-layer ReLU networks but instead of looking at the standard single output regression / classification setting, the authors study multiple outputs. In particular, the dataset X is associated with labels y and another dataset X_aux is associated with labels y_aux, which comprise an auxiliary task. With these, they study the multitask setting and the pretraining and fine-tuning setting. First, they apply several pre-existing results on implicit regularization to derive the implicit regularizer in each of these cases. They then analyze the qualitative implications of these results and run experiments to validate those qualitative analyses.

**Strengths:**

* I really like the style of the paper! Take a theoretical result, extend it to a new setting, understand its implications, and verify them! The implications are not only verified on simplified models but also in more realistic models.
* I view the audience of your paper as a much broader community than just the theory of deep learning community, as there are concrete and possibly algorithmic implications for your findings (i.e., it may be possible to design algorithms that actively incentivize certain kinds of feature learning in different settings via your results).

**Weaknesses:**

A good paper overall. See some questions below.

**Questions:**

* This paper seems somewhat related to the PT + FT setting. Mainly out of curiosity, as I think the settings are a little different, do you see any connections?

Evron, I., Moroshko, E., Buzaglo, G., Khriesh, M., Marjieh, B., Srebro, N. & Soudry, D.. (2023). Continual Learning in Linear Classification on Separable Data. Proceedings of the 40th International Conference on Machine Learning.
* These papers seem related to your feature sparsity and sharing experiments. Can you please discuss how they relate? In particular, do you see the emergence of a similar phenomenon to these works in the intermediate regime?

Lee, S., Goldt, S. & Saxe, A.. (2021). Continual Learning in the Teacher-Student Setup: Impact of Task Similarity. Proceedings of the 38th International Conference on Machine Learning.

Lee, S., Mannelli, S.S., Clopath, C., Goldt, S. & Saxe, A.. (2022). Maslow’s Hammer in Catastrophic Forgetting: Node Re-Use vs. Node Activation. Proceedings of the 39th International Conference on Machine Learning

* While for someone familiar with the literature it is not so necessary, for accessibility to a more general audience, it would be helpful to get a little more discussion on what the settings and results are in [Dai et al], [Woodworth et al], [Azulay et al] and in particular why you need different results for the different settings. As mentioned above, your paper is one for a broad audience, one that is certainly much broader than just the implicit regularization community, who would be the ones most familiar with the subtleties in those above results.

**Limitations:**

Work is self-contained and aims to explain; the limitations of the theory in explaining certain cases are raised and addressed.

---

> ### Author Rebuttal · Authors · 2024-08-06
>
> Thank you very much for your positive feedback and your helpful comments. Below we respond to your questions.
>
> > This paper seems somewhat related to the PT + FT setting. Mainly out of curiosity, as I think the settings are a little different, do you see any connections?
>
> We agree that this paper is related. This paper (which focuses on classification, though see Evron et al. (2022), which focuses on regression) also analyzes sequential learning, focusing on how learning of subsequent tasks affects forgetting of previous tasks (i.e. a continual learning perspective). In contrast, we focus (in our PT+FT analysis) on how learning of a previous task impacts generalization on the subsequent task. Further, the authors analyze a shallow network using only a linear readout, whereas we analyze networks with a hidden layer, enabling us to characterize the impact of feature learning (e.g. giving rise to feature sparsity and nested sparsity biases). Thus, these papers use complementary methods to study a related, but different problem. Future work could both analyze generalization on sequential tasks using the methods established in Evron et al., or use our analysis to analyze the continual learning setup. We find both of these directions quite promising and will add the paper to the Related Work section.
>
> > These papers seem related to your feature sparsity and sharing experiments. Can you please discuss how they relate? In particular, do you see the emergence of a similar phenomenon to these works in the intermediate regime?
>
> There are indeed a number of similarities between these papers and our own investigation. In particular, they also investigate a teacher-student setup (albeit with a different nonlinearity), using two different teachers with various overlaps. Notably, their focus (just like that of Evron et al.) is on forgetting, but they also investigate forward transfer, studying how well the networks can learn novel tasks with various overlaps. In particular, they find that these networks do not always reach optimal training error, which may be due to the online learning setup, the challenging learning landscape caused by the initialization induced by pretraining, and the fact that the student networks have about as many parameters as the teacher. In contrast, we study overparameterized student networks which are able to reach arbitrarily low training error and then investigate their generalization error. Nevertheless, these phenomena may be affected by similar mechanisms: in particular, Lee et al. (2021; 2022) also manipulate the alignment between the two teachers by varying the correlations of their hidden features and find maximal forgetting for intermediate correlations. While we focus on the generalization error on the finetuning task (rather than measuring forgetting on the pretraining task), we expect that in our experiments, the error on the original pretraining dataset after finetuning (i.e. a measure of forgetting) might exhibit a similar non-monotonicity with respect to correlations.
>
> Notably, Lee et al. (2022) consider interleaved replay (similar to MTL in our setup) and find that it performs worse than finetuning (or regularized finetuning) for intermediate task similarity. This is also the case in our studies, where correlated teacher features (corresponding to intermediate task similarity) generally yield worse performance on MTL compared to PT+FT. Indeed, the intuition of Maslow's hammer may transfer to our setting: MTL networks try to re-use the same features on both tasks, which harms their generalization. In contrast, finetuned networks change their pretrained features (which yields forgetting in the setup of Lee et al.), yielding better generalization on the finetuning task.
>
> All in all, these prior works on continual learning provide useful contextualization of our own work. We will add a paragraph on this to the related work section and will further note the similarity discussed in the previous paragraph in section 4.4. Thank you for bringing these papers to our attention!
>
> > While for someone familiar with the literature it is not so necessary, for accessibility to a more general audience, it would be helpful to get a little more discussion on what the settings and results are in [Dai et al], [Woodworth et al], [Azulay et al] and in particular why you need different results for the different settings. As mentioned above, your paper is one for a broad audience, one that is certainly much broader than just the implicit regularization community, who would be the ones most familiar with the subtleties in those above results.
>
> We agree with this point and will add a paragraph to the related work section on the contrast between implicit and explicit regularization, explaining why we are using a mixture of both. Roughly, our summary is: Woodworth et al. and Azulay et al. are able to characterize the implicit regularization induced by gradient descent for diagonal linear networks trained from arbitrary initialization. However, it is technically much more challenging to derive a similar result for multi-output diagonal linear networks, or for ReLU networks of any kind. It is generally easier to characterize the impact of explicit weight regularization, which Dai et al. do for multi-output diagonal linear networks. Our own contributions to the theoretical landscape are (1) spelling out the implications of existing results on implicit regularization in diagonal linear networks, when applied to the PT+FT setting, and (2) characterization of the penalty conferred by explicit regularization on finetuning ReLU networks from arbitrary initialization. Again, we take an explicit regularization perspective in the ReLU case (and validate experimentally that our results are a good description of implicit regularization dynamics), because this is technically much more tractable than characterizing the implicit regularization of ReLU networks.

---

> > ### Comment · Reviewer_969G · 2024-08-11
> > **Thank you!**
> >
> > Thanks for engaging with my suggestions. I hope you found the literature suggestions valuable. I appreciate the discussion you have provided here. To reiterate, I really like your paper!

---

> > > ### Author Response · Authors · 2024-08-13
> > >
> > > Thank you again for your helpful review, your positive assessment of our work, and your literature suggestions!

---

### Author Rebuttal · Authors · 2024-08-06

We thank the reviewers for their helpful feedback and comments. In particular, the reviewers have suggested a number of relevant papers that we will add to the related work section. In addition, they have provided valuable feedback on clarity, which we will take into account in revising the manuscript (see specific responses to reviewers). We respond to the individual reviews below.

---

### Decision · Program_Chairs · 2024-09-25

**Decision:**

Accept (poster)

**Comment:**

The main contribution of this paper is a theoretical study of the implicit regularization in multi-task learning, comparing it to a pre-training and fine-tuning schedule.  The authors study a simple toy model of diagonal linear network, showing that MTL and PT+FT display different implicit regularization, where PT+FT benefits from features correlated between the main and auxiliary task.

While there were some concerns regarding the applicability of the theoretical results derived for the simple toy model studied in the paper, reviewers agree that the results in the paper are novel and that the setting studied in the paper and the implication of the results are very interesting. I therefore recommend that this paper is accepted.